# Lactate dehydrogenase A regulates tumor-macrophage symbiosis to promote glioblastoma progression

Fatima Khan[1], Yiyun Lin [2,3], Heba Ali [1], Lizhi Pang[1], Madeline Dunterman [1], Wen-Hao Hsu[3,4], Katie Frenis [5,6], R. Grant Rowe[5,6,7], Derek A. Wainwright [1], Kathleen McCortney [1], Leah K. Billingham[1], Jason Miska [1], Craig Horbinski [1,8], Maciej S. Lesniak [1] & Peiwen Chen [1]✉

Abundant macrophage infiltration and altered tumor metabolism are two key hallmarks of glioblastoma. By screening a cluster of metabolic small-molecule compounds, we show that inhibiting glioblastoma cell glycolysis impairs macrophage migration and lactate dehydrogenase inhibitor stiripentol emerges as the top hit. Combined profiling and functional studies demonstrate that lactate dehydrogenase A (LDHA)-directed extracellular signal-regulated kinase (ERK) pathway activates yes-associated protein 1 (YAP1)/signal transducer and activator of transcription 3 (STAT3) transcriptional co-activators in glioblastoma cells to upregulate C-C motif chemokine ligand 2 (CCL2) and CCL7, which recruit macrophages into the tumor microenvironment. Reciprocally, infiltrating macrophages produce LDHA-containing extracellular vesicles to promote glioblastoma cell glycolysis, proliferation, and survival. Genetic and pharmacological inhibition of LDHA-mediated tumor-macrophage symbiosis markedly suppresses tumor progression and macrophage infiltration in glioblastoma mouse models. Analysis of tumor and plasma samples of glioblastoma patients confirms that LDHA and its downstream signals are potential biomarkers correlating positively with macrophage density. Thus, LDHA-mediated tumor-macrophage symbiosis provides therapeutic targets for glioblastoma.

Glioblastoma is a devastating brain tumor in human adults with a median survival averaging 15-20 months following initial diagnosis[1,2]. Unfortunately, current therapies have failed to improve the survival of glioblastoma patients meaningfully over the last four decades[3–7]. Due to glioblastoma cell heterogeneity and genetic instability, clinical trials for targeted therapies (e.g., therapies targeting receptor tyrosine kinase signaling) have also failed to improve glioblastoma patient outcomes[8,9]. There is an increasing recognition that the signaling from glioblastoma cells not only impacts cancer cell biology, but also regulates the biology (e.g., recruitment and activation) of immune cells in the tumor microenvironment (TME), thus inducing a tumor-immune cell symbiotic interaction[5,6]. Among the TME, tumor-associated

[1]Department of Neurological Surgery, Lou and Jean Malnati Brain Tumor Institute, Robert H Lurie Comprehensive Cancer Center, Feinberg School of Medicine, Northwestern University, Chicago, IL 60611, USA. [2]Department of Genetics, The University of Texas MD Anderson Cancer Center, Houston, TX, USA. [3]UTHealth Graduate School of Biomedical Sciences, The University of Texas MD Anderson Cancer Center, Houston, TX, USA. [4]Department of Cancer Biology, The University of Texas MD Anderson Cancer Center, Houston, TX 77054, USA. [5]Department of Hematology, Boston Children's Hospital, Boston, MA 02115, USA. [6]Harvard Medical School, Boston, MA 02115, USA. [7]Dana-Farber Boston Children's Cancer and Blood Disorders Center, Boston, MA 02115, USA. [8]Department of Pathology, Northwestern University Feinberg School of Medicine, Chicago, IL 60611, USA. ✉e-mail: peiwen.chen@northwestern.edu

macrophages and microglia (TAMs) are the largest and most prominent population of immune cells, which account for up to 50% of total live cells in glioblastoma tumor mass[10,11]. Our recent studies have demonstrated that PTEN–yes-associated protein 1 (YAP1)–lysyl oxidase (LOX) signaling in glioblastoma cells, and CLOCK–olfactomedin like 3–legumain and tissue factor pathway inhibitor 2 signaling in glioblastoma stem cells (GSCs) are the key drivers for the infiltration of macrophages and microglia, respectively, which, in turn, promote tumor growth and immunosuppression in glioblastoma[3,12–14]. Such studies highlight the opportunity of identifying the key signals that establish symbiotic interactions between cancer cells and the TME, thus inducing a pro-tumor and immunosuppressive environment for glioblastoma tumorigenesis.

Metabolic reprogramming enables cancer cell growth and proliferation, which is recognized as a prominent hallmark of cancer[15]. Interestingly, recent studies have shown that metabolic reprogramming (such as the regulation of glucose, lipid, tryptophan, and $NAD^+$ metabolism) in cancer cells evades anti-tumor immunity by suppressing lymphocytes[16–18] and recruiting immunosuppressive myeloid cells, including macrophages[19–21]. These findings gain added significance as myeloid cells (e.g., macrophages), lymphocytes, and glioblastoma cells, as well as their symbiotic interactions, are critical for affecting tumor growth and immunotherapy resistance[5,6,8,22]. Encouraged by their functional significance[6], a large body of pharmacological tools has been proposed to target these symbiotic interactions in glioblastoma mouse models[5]. However, certain challenges remain, such as the blood-brain barrier (BBB) that can limit drug delivery into the glioblastoma TME. This creates difficulties in regard to translating the preclinical findings into the clinic[5]. Together, these insights prompted us to conduct a screen of metabolic and brain-penetrant small-molecule compounds that may inhibit glioblastoma cell-induced macrophage infiltration. In this screen, lactate dehydrogenase (LDH) inhibitor stiripentol emerged as the top hit.

LDH is a key player in glucose metabolism that regulates the conversion between pyruvate and lactate. LDH is comprised of two major subunits (e.g., LDHA and LDHB) with LDHA converting pyruvate to lactate in anaerobic conditions and LDHB favoring lactate to pyruvate in the presence of oxygen[23]. However, most cancer cells use aerobic glycolysis (also known as "Warburg effect") to maintain their tumor potential even in the presence of oxygen and produce high levels of lactate[24,25]. Increasing evidence has shown that LDHA-mediated glycolysis promotes glioblastoma cell proliferation and survival and induces resistance to radiotherapy and chemotherapy[26–29]. However, the potential link between immune cells and LDHA-mediated tumor glycolysis in glioblastoma has not been established.

In this work, we elucidate the essential role and molecular mechanism of glioblastoma cell LDHA in promoting macrophage infiltration into the TME and reveal the co-dependencies for macrophage-derived extracellular vesicles (EVs) in supporting glioblastoma cell glycolysis, growth, and survival. Preclinical trials in glioblastoma mouse models, followed by clinical-pathological validations using patient tumor and plasma samples, point to LDHA and its downstream signals as promising therapeutic targets for glioblastoma.

## Results

### Glioblastoma cell glycolysis promotes macrophage migration

The metabolism signature and immune score in The Cancer Genome Atlas (TCGA) glioblastoma tumors have been defined based on gene expression data to infer the levels of tumor metabolism[30] and immune cell populations[31], respectively. To identify the potential connection between tumor metabolism and immunity that might influence glioblastoma tumor biology, GBM patient survival, and correlation analysis between tumor metabolism and immune signatures were performed. We found that high tumor metabolism signature was correlated with poor outcomes (Fig. 1a) and correlated positively with immune score (Fig. 1b). These findings aligned with the Gene Set Enrichment Analysis (GSEA) on hallmark pathways, Gene Ontology Enrichment Analysis (GOEA) on the sub-ontologies of Biological Process, and KEGG Enrichment Analysis showing prominent representations of cytokine and chemokine signatures, immune response networks, and leukocyte and myeloid cell signatures in metabolism-high TCGA glioblastoma patient tumors compared to metabolism-low patient tumors (Table S1). To identify specific immune cells linked to tumor metabolism in glioblastoma, we audited the TCGA glioblastoma patient tumors for 18 types of immune cells using validated gene set signatures[3,12,13,32,33]. As a result, macrophage and monocyte were identified as the top immune cell types correlating positively with the metabolism signature (Fig. 1c). Conversely, CD8+ activated T cells showed a negative correlation with the metabolism signature (Fig. 1c). Together, these findings suggest a connection between tumor metabolism and macrophage/monocyte infiltration in glioblastoma patient tumors.

Given the importance of macrophages in glioblastoma progression[8], we hypothesized that pharmacological inhibition of tumor metabolism-induced macrophage infiltration is a promising therapeutic strategy[5]. We selected a cluster of 55 brain-penetrant small-molecule compounds with metabolic reprogramming functions (Table S2) and performed a screen focusing on macrophage migration using conditioned media (CM) from CT2A cells treated with or without these compounds at 10 μM. This screen resulted in identification of 24 compounds that significantly inhibited CT2A CM-induced macrophage migration (Fig. 1d and Supplementary Fig. S1a). Next, we performed a second round of screen with these 24 compounds at a lower concentration (5 μM) and found that 7 (stiripentol, lopinavir, ofloxacin, vorasidenib, IDH889, progesterone, and IOX4) of them impaired CT2A CM-induced macrophage migration (Fig. 1e and Supplementary Fig. S1b). Consistently, the LDH inhibitor stiripentol showed the strongest effect in these two rounds of screens, which led us to hypothesize that glioblastoma cell glycolysis is essential for macrophage infiltration. To confirm it, we analyzed the single-cell RNA sequencing (scRNA-seq) data from 44 fragments of tumor tissues of 18 glioma patients, including 2 low-grade gliomas (LGG), 11 newly diagnosed glioblastoma (ndGBM), and 5 recurrent glioblastoma (rGBM)[34]. Specifically, glioblastoma tumors containing tumor cells (Fig. 1f) and tumor-infiltrating myeloid cells (Fig. 1g) were analyzed. The glycolysis hallmark signature[35] was highly expressed in glioblastoma cells (Fig. 1h), which correlated positively with the abundance of macrophages and monocytes (Fig. 1i) and negatively with the level of microglia, but did not show a significant correlation with dendritic cells (DCs) (Supplementary Fig. S2a). Similarly, bioinformatics analyses in TCGA glioblastoma tumors demonstrated that high glycolysis signature correlated with increased immune score (Supplementary Fig. S2b); prominent representations of immune response networks, cytokine and chemokine signatures, as well as leukocyte and myeloid cell migration signatures (Table S3); and increased macrophages, monocytes, and to a lesser extent, DCs, and decreased microglia (Supplementary Fig. S2c). Finally, glycolysis signature, but not other metabolism-related hallmark signatures, was enriched in glioblastoma patient tumors compared to normal brain tissues (Supplementary Fig. S2d, e), and was increased in glioma cells of glioblastoma compared to LGG (Supplementary Fig. S2f).

To biologically validate the role of glycolysis in triggering macrophage infiltration, we treated mouse glioblastoma cells (e.g., CT2A and GL261) and glioblastoma patient-derived GSC272 with a glycolysis inhibitor 2-deoxy-D-glucose (2-DG)[21], which can inhibit glycolysis as shown by reduced extracellular acidification rate (ECAR) and lactate production (Supplementary Fig. S2g-j). As expected, CM from 2-DG-treated glioblastoma cells and GSCs reduced the migration of macrophages, including mouse Raw264.7 macrophages (Fig. 1j-l and Supplementary

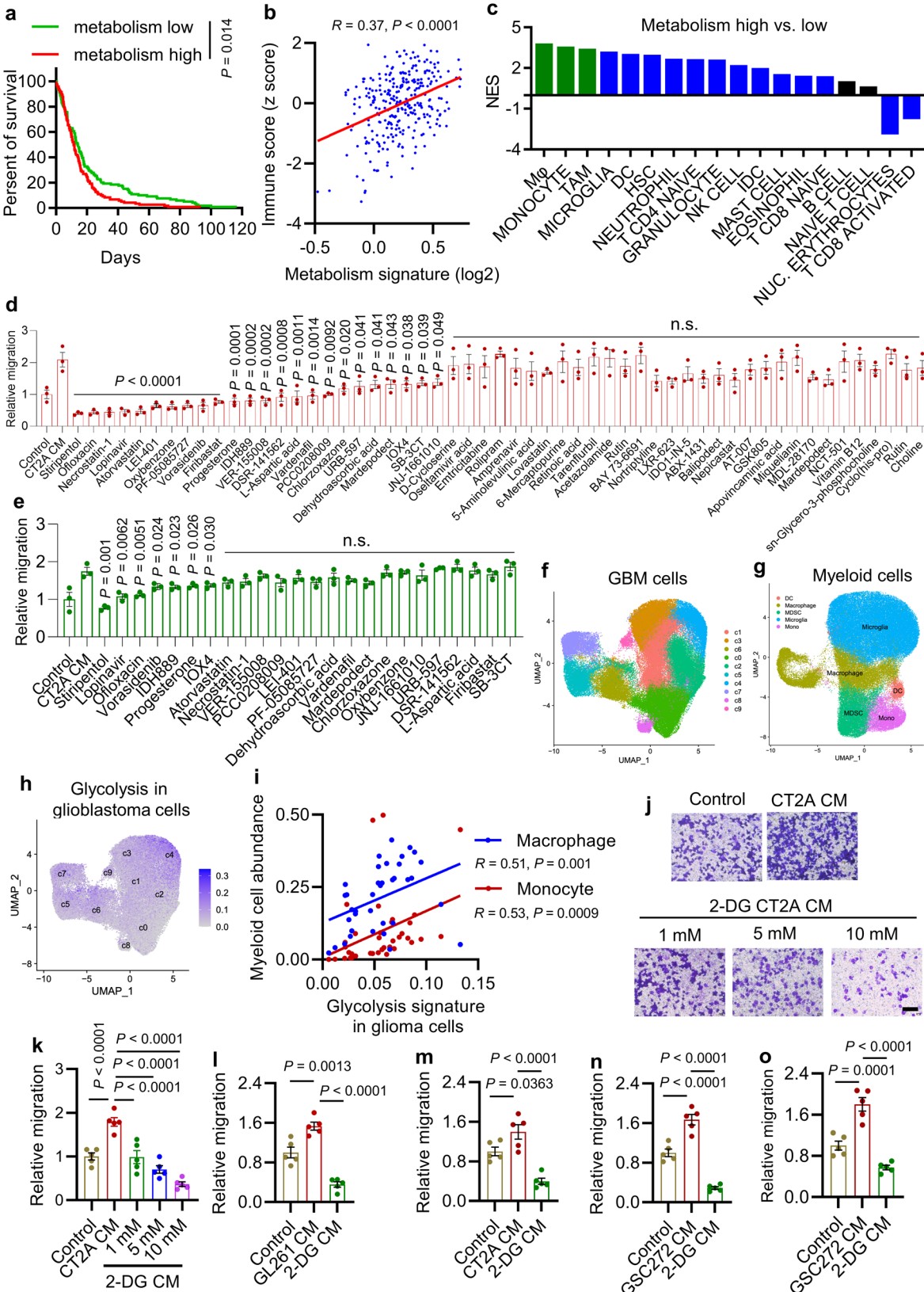

Fig. 2k), primary mouse bone marrow-derived macrophages (BMDMs) (Fig. 1m and Supplementary Fig. S2l), human THP-1 macrophages (Fig. 1n and Supplementary Fig. S2m), and primary human BMDMs (Fig. 1o and Supplementary Fig. S2n), relative to the CM from control cells. Together, these findings suggest a critical role of glioblastoma cell glycolysis in regulating macrophage infiltration.

## Glioblastoma cell LDHA promotes macrophage infiltration

To determine the molecular basis of tumor glycolysis in support of macrophage infiltration, we examined the connection between the expression of key glycolysis and tricarboxylic acid (TCA) cycle enzymes (e.g., HK1, HK2, HK3, PGM1, PGM2, LDHA, LDHB, MDH1, MDH2, FH, SDHA, SUCLA2, OGDH, IDH3A, IDH3B, IDH3G, CS, and

**Fig. 1 | Glioblastoma cell glycolysis promotes macrophage migration. a** Kaplan-Meier survival curves of GBM patients relative to high ($n = 119$) and low ($n = 119$) metabolism signature[30]. **b** The correlation analysis between metabolism signature and immune score[31] in IDH-WT glioblastoma patient tumors ($n = 300$). **c** GSEA analysis for distinct types of immune cells in metabolism signature-high ($n = 119$) and -low ($n = 119$) patient tumors from TCGA glioblastoma dataset. Green and blue bars indicate the signatures that are significantly enriched metabolism signature-high patient tumors (FDR < 0.25). **d** Quantification of relative migration of Raw264.7 macrophages following stimulation with conditioned media (CM) from CT2A cells treated with or without a cluster of 55 brain-penetrant small-molecule compounds with metabolic reprogramming functions at 10 μM ($n = 3$ biological replicates). n.s., not significant. **e** Quantification of relative migration of Raw264.7 macrophages following stimulation with CM from CT2A cells treated with or without above identified 24 compounds at 5 μM ($n = 3$ independent samples). n.s., not significant. **f, g** UMAP dimensional reduction of single glioma cells **f** and myeloid cells **g** from tumors of 18 glioma patients, including 2 low-grade gliomas (LGG) and 16 glioblastoma[34]. **h** Expression pattern representing single-cell gene expression of glycolysis signature in glioblastoma cells. **i** The correlation analysis between the glycolysis signature in glioblastoma cells and the abundance of macrophages and monocytes in tumors based on single-cell RNA sequencing data[34]. Each dot represents one glioblastoma patient tumor. **j, k** Representative images **j** and quantification **k** of relative migration of Raw264.7 macrophages from a transwell analysis following stimulation with CM from CT2A cells treated with or without glycolysis inhibitor 2-Deoxy-d-glucose (2-DG) at indicated concentrations ($n = 5$ independent samples). Scale bar, 100 μm. **l, m** Quantification of relative migration of Raw264.7 macrophages **l** and primary mouse bone-marrow-derived macrophages **m** following stimulation with CM from GL261 and CT2A cells, respectively, treated with or without 2-DG at 10 mM ($n = 5$ independent samples). **n, o** Quantification of relative migration of THP-1 macrophages **n** and primary human bone-marrow-derived macrophages **o** following stimulation with CM from GSC272 treated with or without 2-DG at 10 mM ($n = 5$ independent samples). The experiments for **e** and **j–o** were independently repeated at least two times. Data presented as mean ± SEM. Statistical analyses were determined by log-rank test **a**, Pearson's correlation test **b, i** and one-way ANOVA test **d, e, k, l, m, n, o**. Source data are provided as a Source Data file.

ACO1) with patient survival, immune score, and macrophage signature in TCGA glioblastoma patient tumors. Following these analyses, *LDHA* was identified as the only gene that correlated negatively with patient survival and positively with immune score and macrophage signature (Supplementary Fig. S3a). Analysis of the scRNA-seq data[34] also identified *LDHA* as the only gene that was increased in glioma cells of glioblastoma (including ndGBM and rGBM) compared to LGG (Fig. 2a) and correlated positively with the abundance of macrophages and monocytes in glioblastoma patient tumors (Fig. 2b). Next, GSEA on hallmark pathways with the RNA-Seq profiling data from CT2A cells treated with or without a LDHA specific inhibitor FX11 demonstrated that FX11-treated cells displayed an impaired representation of immune response networks including interferon alpha and gamma responses, and inflammatory response (Fig. 2c). Similarly, *LDHA*-high glioblastoma patient tumors showed prominent representations of leukocyte and myeloid cell migration signatures, immune response networks, and cytokine and chemokine signatures (Table S4). Finally, GSEA on distinct immune cell signatures confirmed that macrophage and monocyte were the top immune cell types enriched in *LDHA*-high TCGA glioblastoma patient tumors (Supplementary Fig. S3b).

To further confirm the relevance of LDHA-mediated tumor glycolysis in promoting macrophage infiltration, we conducted shRNA-mediated LDHA depletion (sh*Ldha*) in glioblastoma cells, such as CT2A and GL261 (Fig. 2d), or treated them and GSCs with LDHA inhibitors (e.g., stiripentol and FX11). As expected, these modifications and treatments reduced lactate levels (Supplementary Fig. S3c–g) and ECAR (Supplementary Fig. S3h, i). Furthermore, CM from LDHA-depleted CT2A and GL261 cells reduced macrophage migration relative to CM from shRNA control cells (Fig. 2e–g and Supplementary Fig. S4a). Similarly, CM from stiripentol-treated CT2A cells (Fig. 2h, i and Supplementary Fig. S4b, c), GL261 cells (Fig. 2j and Supplementary Fig. S4d), 005 GSCs, a GSC line isolated from tumors with lentiviral transduction of brains with H-Ras and AKT in *Trp53*[+/-] mice[36,37] (Supplementary Fig. 4e, f), and GSC272 (Fig. 2k, l and Supplementary Fig. S4g, h), induced significantly less migration of macrophages (including Raw264.7 macrophages, primary mouse BMDMs, THP-1 macrophages, and primary human BMDMs) than CM from untreated cells. In addition, CM from FX11-treated CT2A cells (Fig. 2m and Supplementary Fig. S4i), GL261 cells (Fig. 2n and Supplementary Fig. S4j), and GSC272 (Fig. 2o, p and Supplementary Fig. S4k, l) showed similar macrophage migration inhibitory effect. Finally, this phenomenon was reinforced by a scratch assay showing that CT2A CM-induced macrophage migration was impaired when glioblastoma cell LDHA was inhibited genetically and pharmacologically (Supplementary Fig. S4m–p). Together, these findings support a pivotal role of glioblastoma cell/GSC LDHA in triggering macrophage infiltration into the glioblastoma TME.

## Glioblastoma cell LDHA promotes macrophage infiltration via upregulating CCL2 and CCL7

GSEA on KEGG pathways of CT2A cells with FX11 treatment versus control exhibited a prominent reduction of signatures related to chemokine and cytokine-cytokine receptor interaction (Fig. 3a), suggesting that LDHA in glioblastoma cells may regulate the expression of chemokines and cytokines. To elucidate such chemokines and/or cytokines governing macrophage recruitment in LDHA-high glioblastoma cells, we examined putative factors exhibiting a ≥ 2.0-fold change in CT2A cells (FX11 treatment versus control) and TCGA glioblastoma patient tumors (*LDHA*-high versus -low) using a human secreted protein dataset[38]. This analysis led to identification of eleven genes (e.g., *CCL2, CCL7, IL1B, IL1RAP, IL1RN, MMP9, NPY, PLAU, PROS1, S100A8* and *SLPI*) encoding secreted proteins that were upregulated in *LDHA*-high patient tumors compared to *LDHA*-low tumors and downregulated by LDHA inhibitor FX11 treatment in CT2A cells (Fig. 3b, c). To reveal the importance of these genes in glioblastoma tumor biology, we conducted bioinformatics analyses in TCGA glioblastoma tumors showing that the expression of most of these genes (except for *NPY*) correlated positively with macrophage signature, but only *CCL2, CCL7, IL1RAP, PLAU,* and *S100A8* correlated negatively with patient survival (Supplementary Fig. S5a). RT-qPCR demonstrated a decreased expression of *Ccl2, Ccl7, Plau,* and *S100a8*, but not *Il1rap*, in CT2A and GL261 cells upon the treatment with LDHA inhibitor FX11 (Fig. 3d and Supplementary Fig. S5b). Reduced expression of *Ccl2, Ccl7, Plau,* and *S100a8*, was further confirmed by additional pharmacological (using LDHA inhibitor stiripentol) and genetic (using sh*Ldha*) strategies in CT2A cells (Fig. 3e, f) and GL261 cells (Supplementary Fig. S5c, d). To validate the capacity of CCL2, CCL7, PLAU, and S100A8 functioning as macrophage chemoattractants, we performed transwell migration assay showing that recombinant CCL2 and CCL7, but not PLAU and S100A8, protein-supplemented media increased the migration of Raw264.7 macrophages (Fig. 3g, h). Similar experiments in human GSC272 demonstrated that LDHA inhibitor stiripentol treatment reduced CCL2 and CCL7 expression (Fig. 3i, j) and secretion (Fig. 3k, l). Conversely, recombinant LDHA protein treatment increased the expression of CCL2 and CCL7 in both mouse CT2A cells and human GSC272 and rescued the impaired CCL2 and CCL7 levels in sh*Ldha* CT2A cells (Supplementary Fig. S5e–h). Consistent with the data from mouse Raw264.7 macrophages, recombinant CCL2 and CCL7 protein-supplemented media increased the migration of human THP-1 macrophages (Fig. 3m and Supplementary Fig. S5i). More importantly, the impaired macrophage migration induced by CM from sh*Ldha* CT2A cells was prevented by the treatment with recombinant CCL2 and CCL7 proteins (Supplementary Fig. S5j–m). The GOEA on the sub-ontologies of Biological Process and Molecular Function, and KEGG Enrichment Analysis in TCGA glioblastoma patient

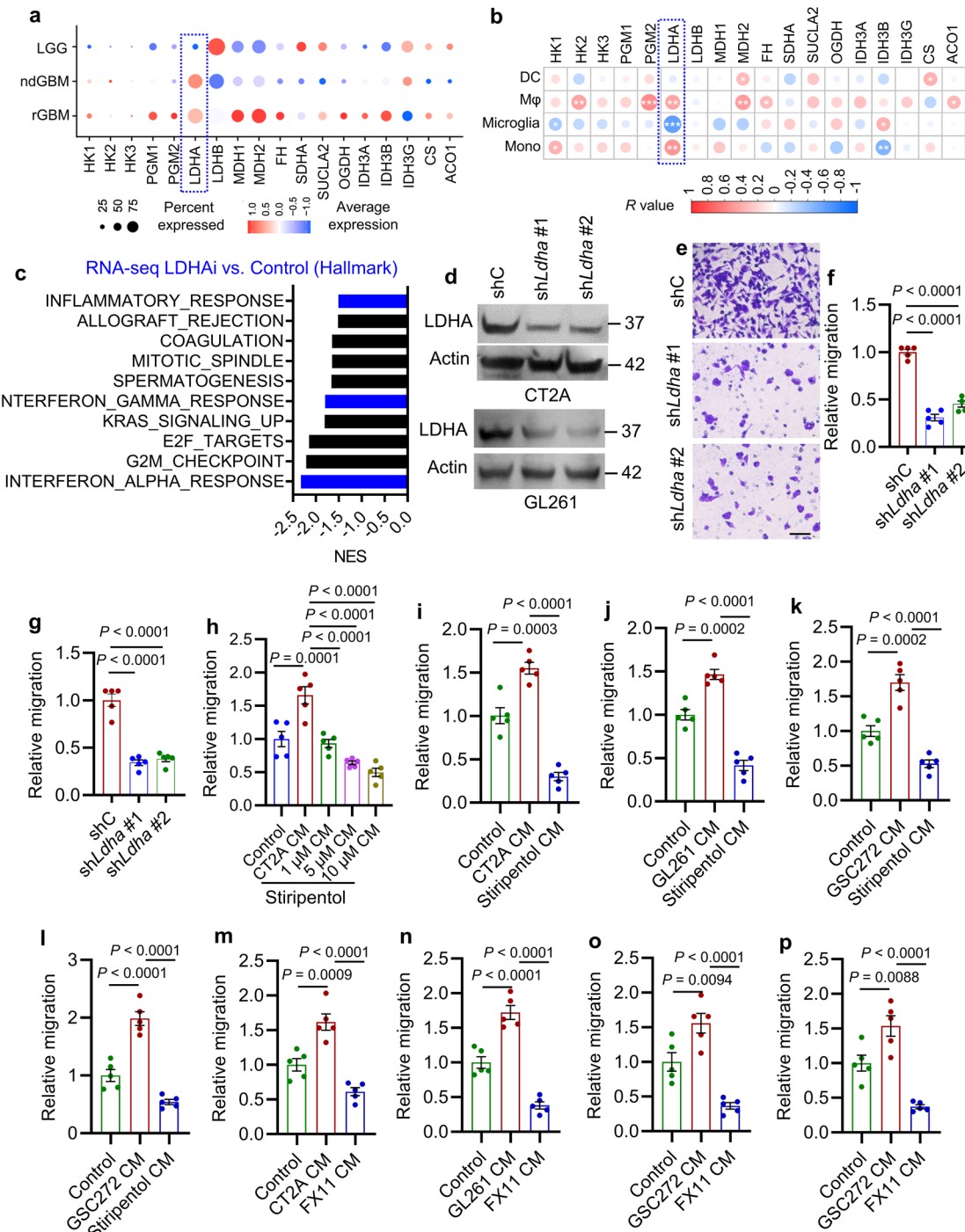

tumors also demonstrated that the migration of leukocytes and/or myeloid cells and the activity of chemokines and cytokines were the top CCL2- and CCL7-regulated processes (Tables S5 and S6).

To further confirm the role of glioblastoma cell CCL2 and CCL7 in macrophage infiltration, we first employed shRNAs to deplete CCL2 and CCL7 in CT2A and GL261 cells. As expected, CM from CT2A and GL261 cells expressing sh*Ccl2* (Fig. 3n, o and Supplementary Fig. S6a–d) and sh*Ccl7* (Fig. 3p, q and Supplementary Fig. S6e–h) induced significantly less macrophage migration than CM from shRNA control (shC) cells. Next, we depleted CCL2 and CCL7 in GSC272 and confirmed that CM from GSC272 expressing sh*CCL2* (Fig. 3r, s and Supplementary Fig. S6i, j) and sh*CCL7* (Fig. 3t, u and Supplementary Fig. S6k, l) inhibited the migration of THP-1 macrophages and primary human BMDMs compared to CM from shC cells. In summary, these

results reinforce that the expression of CCL2 and CCL7 in glioblastoma cells/GSCs is regulated by LDHA and that glioblastoma cell CCL2 and CCL7 function as potent macrophage chemoattractants.

## YAP1 and STAT3 transcriptional co-activators regulate LDHA-induced CCL2 and CCL7 expression in glioblastoma cells

To explore how LDHA regulates CCL2 and CCL7 expression, GSEA was utilized to catalog oncogenic signaling pathways modulated by LDHA in glioblastoma cells (LDHA inhibitor FX11 versus control) and TCGA glioblastoma patient tumors (*LDHA*-high versus *LDHA*-low). As a result, 35 overlapping pathways were identified, which include transcription factors (e.g., YAP1, ATF2, HOXA9, LEF1, and NRL), signaling pathways (e.g., YAP1, JNK/STAT, AKT/mTOR, Raf/ERK, and STK33), epigenetic factors (e.g., EED and EZH2), tumor suppressor genes and oncogenes

**Fig. 2 | Glioblastoma cell LDHA promotes macrophage migration. a** Expression of key glycolysis and TCA cycle enzymes in glioma cells of low-grade gliomas (LGG), newly diagnosed glioblastoma (ndGBM), and recurrent glioblastoma (rGBM) based on single-cell RNA sequencing data[34]. The percent and average expressions are shown. **b** The correlation between key glycolysis and TCA cycle enzymes in glioblastoma cells and myeloid cells, including dendritic cells (DCs), macrophages (Mφ), microglia, and monocytes (Mono) from glioblastoma patient tumors[34]. Red signal indicates positive correlation and blue signal denotes a negative correlation. \*$P < 0.05$, \*\* $P < 0.01$, \*\*\*$P < 0.001$. **c** RNA-seq experiments and GSEA analysis in LDHA inhibitor FX11-treated and control CT2A cells. Top ten FX11-downregulated hallmark pathways are shown. Blue bars indicate the signatures relating to immune response. **d** Immunoblots of LDHA in cell lysates of CT2A and GL261 cells expressing shRNA control (shC) and *Ldha* shRNAs (sh*Ldha*). The experiments were independently repeated at least three times. **e, f** Representative images **e** and quantification **f** of relative migration of Raw264.7 macrophages from a transwell analysis following stimulation with CM from CT2A cells expressing shC and sh*Ldha*. Scale bar, 100 µm. *n* = 5 independent samples. **g** Quantification of relative migration of Raw264.7 macrophages following stimulation with CM from shC and sh*Ldha* GL261 cells. *n* = 5 independent samples. **h** Quantification of relative migration of Raw264.7 macrophages following stimulation with CM from CT2A cells treated with or without stiripentol. *n* = 5 independent samples. **i, j** Quantification of relative migration of primary mouse bone-marrow-derived macrophages (BMDMs, **i** and Raw264.7 macrophages **j** following stimulation with CM from CT2A and GL261 cells, respectively, treated with or without stiripentol (10 µM). *n* = 5 independent samples. **k, l** Quantification of relative migration of THP-1 macrophages **k** and primary human BMDMs **l** following stimulation with CM from GSC272 treated with or without stiripentol (10 µM). *n* = 5 independent samples. **m, n** Quantification of relative migration of Raw264.7 macrophages following stimulation with CM from CT2A **m** or GL261 **n** cells treated with or without FX11 (8 µM). *n* = 5 independent samples. **o, p** Quantification of relative migration of THP-1 macrophages (**o**) and primary human BMDMs (**p**) following stimulation with CM from GSC272 treated with or without FX11 (8 µM). *n* = 5 independent samples. The experiments for (**e**–**p**) were independently repeated at least two times. Data presented as mean ± SEM. Statistical analyses were determined by Pearson's correlation test (**b**) and one-way ANOVA test (**f**–**p**). Source data are provided as a Source Data file.

(e.g., *KRAS*, *TP53*, *RB*, and *SNF5*), and others (e.g., IL2, RPS14, VEGFA, and WNT1) (Fig. 4a). By analyzing RNA-Seq data from CT2A cells focusing on above identified factors, we found that the expression of *Hoxa9*, *Yap1*, *Eed*, *Ezh2*, and *Trp53* was downregulated by LDHA inhibitor FX11 treatment (Fig. 4b), which was confirmed by RT-qPCR analysis in both CT2A and GL261 cells (Fig. 4c and Supplementary Fig. S7a). Further studies on CT2A and GL261 cells treated with stiripentol (Fig. 4d and Supplementary Fig. S7b) or expressing sh*Ldha* (Fig. 4e, f) demonstrated that LDHA inhibition downregulated the expression of *Hoxa9* and *Yap1*, but had no effect on *Eed* and *Ezh2*. Next, we aimed to confirm whether YAP1, JNK/STAT, AKT/mTOR, Raf/ERK, and STK33 pathways are regulated by LDHA in glioblastoma cells. Western blotting demonstrated that shRNA-mediated depletion of LDHA or LDHA inhibitor (e.g., FX11 and stiripentol) treatment in CT2A cells and GSC272 significantly inhibited Phospho-ERK (P-ERK), YAP1, and P-STAT3 (Fig. 4g–i), but did not affect STK33, P-AKT, and P-STAT6 (Supplementary Fig. S7c, d). Moreover, the decreased P-ERK, YAP1, and P-STAT3 was confirmed in GL261 cells expressing sh*Ldha* or treated with LDHA inhibitors FX11 and stiripentol (Supplementary Fig. S7e–g). Finally, we treated mouse CT2A and GL261 cells and human GSC272 with ERK inhibitor ravoxertinib and found that such treatments significantly reduced YAP1, P-STAT3, and HOXA9 (Supplementary Fig. S7h–k). Together, these findings suggest that LDHA-directed ERK pathway regulates HOXA9, YAP1, and STAT3 transcription factors and/or signaling pathways in glioblastoma cells and GSCs.

To investigate the potential functional relevance of HOXA9, YAP1, and STAT3 in regulating CCL2 and CCL7 expression and macrophage infiltration, bioinformatics analyses in TCGA glioblastoma patient tumors were performed. As a result, we found that the expression of *YAP1* and *STAT3*, but not *HOXA9* and *TP53*, correlated positively with *CCL2*, *CCL7*, and macrophage signature (Supplementary Fig. S7l). Then, CT2A cells, GL261 cells, and GSC272 were treated with YAP1-TEAD interaction inhibitor verteporfin[39] and STAT3 inhibitor WP1066. The results of these experiments demonstrated that verteporfin treatment reduced P-STAT3, and, reciprocally, WP1066 treatment impaired YAP1 expression at both mRNA and protein levels (Fig. 4j, k). Moreover, the nuclear localization of STAT3 was reduced when CT2A cells harboring sh*Ldha* or treated with stiripentol and verteporfin (Supplementary Fig. S7m, n). Similarly, depletion of LDHA or treatment with stiripentol and WP1066 in CT2A cells reduced the nuclear localization of YAP1 (Supplementary Fig. S7o, p). These findings suggest that LDHA-regulated YAP1 and STAT3 are transcriptional co-activators[40], prompting us to investigate the role of YAP1 and STAT3 in transcriptional regulation of *CCL2* and *CCL7* in glioblastoma cells. Correspondingly, we observed specific YAP1 and STAT3 binding to the *Ccl2* and *Ccl7* promoters in CT2A cells, which was reduced upon LDHA depletion (Fig. 4l, m). Moreover, pharmacological treatment with verteporfin and WP1066 in CT2A and GL261 cells repressed *Ccl2* and *Ccl7* expression (Fig. 4n, o). To further investigate whether LDHA-regulated lactate contributes to this process, we treated LDHA-depleted glioblastoma cells with lactate and found that this treatment rescued the impaired signaling of P-ERK, YAP1, P-STAT3, CCL2, and CCL7 in sh*Ldha* CT2A cells (Supplementary Fig. S7q–s). Together, these findings suggest that YAP1 and STAT3 transcriptional co-activators contribute to LDHA/lactate–ERK axis-dependent CCL2 and CCL7 expression in glioblastoma cells.

## Macrophage-derived LDHA-containing EVs promote tumor growth and activate the ERK-YAP1/STAT3-CCL2/CCL7 axis in glioblastoma cells

Once infiltrating into the glioblastoma TME, macrophages are educated to promote glioblastoma progression by secreting distinct factors and EVs[8]. To mimic this process, we first utilized glioblastoma cell CM to educate macrophages (hereafter such educated macrophages are referred to as EMφ), and then examined the role of CM from EMφ on glioblastoma cells. As a result, we found that EMφ CM promoted LDHA expression in CT2A and GL261 cells (Fig. 5a, b), prompting a speculation that TAMs may support glioblastoma cell growth and survival via upregulating LDHA. To confirm the role of LDHA in glioblastoma cell biology, we performed cell cycle, apoptosis, and proliferation analyses in glioblastoma cells with or without LDHA inhibition. We found that CT2A cells expressing sh*Ldha* or treated with LDHA inhibitors (e.g., isosafrole, FX11 or stiripentol) displayed decreased G1 and upregulated G2–M fractions (Supplementary Fig. S8a–d), upregulated apoptosis (Supplementary Fig. S8e–h), and reduced proliferation (Supplementary Fig. S8i–l).

To reveal how TAMs upregulate LDHA in glioblastoma cells, we depleted LDHA using shRNAs (Supplementary Fig. S9a) and inhibited LDHA using FX-11 in EMφ. Surprisingly, we noticed that LDHA inhibition in macrophages abolished EMφ CM-induced LDHA upregulation in glioblastoma cells (Fig. 5a, b), suggesting a potential for LDHA delivery from EMφ to glioblastoma cells. scRNA-seq data analysis on tumors from a cohort of four glioblastoma patients[41] demonstrated that LDHA was highly expressed in both glioblastoma cells and CD68+CX3CR1- macrophages, but not in CD68+CX3CR1+ microglia (Supplementary Fig. S9b–e). As expected, genetic and pharmacological inhibition of LDHA in macrophages reduced glycolysis as shown by the impaired lactate production (Supplementary Fig. S9f, g) and ECAR (Supplementary Fig. S9h, i). To investigate whether LDHA could be delivered from macrophages into glioblastoma cells via EVs, we treated EMφ with GW4869 (an EV biogenesis and release inhibitor) and found that this treatment abolished EMφ CM-induced LDHA

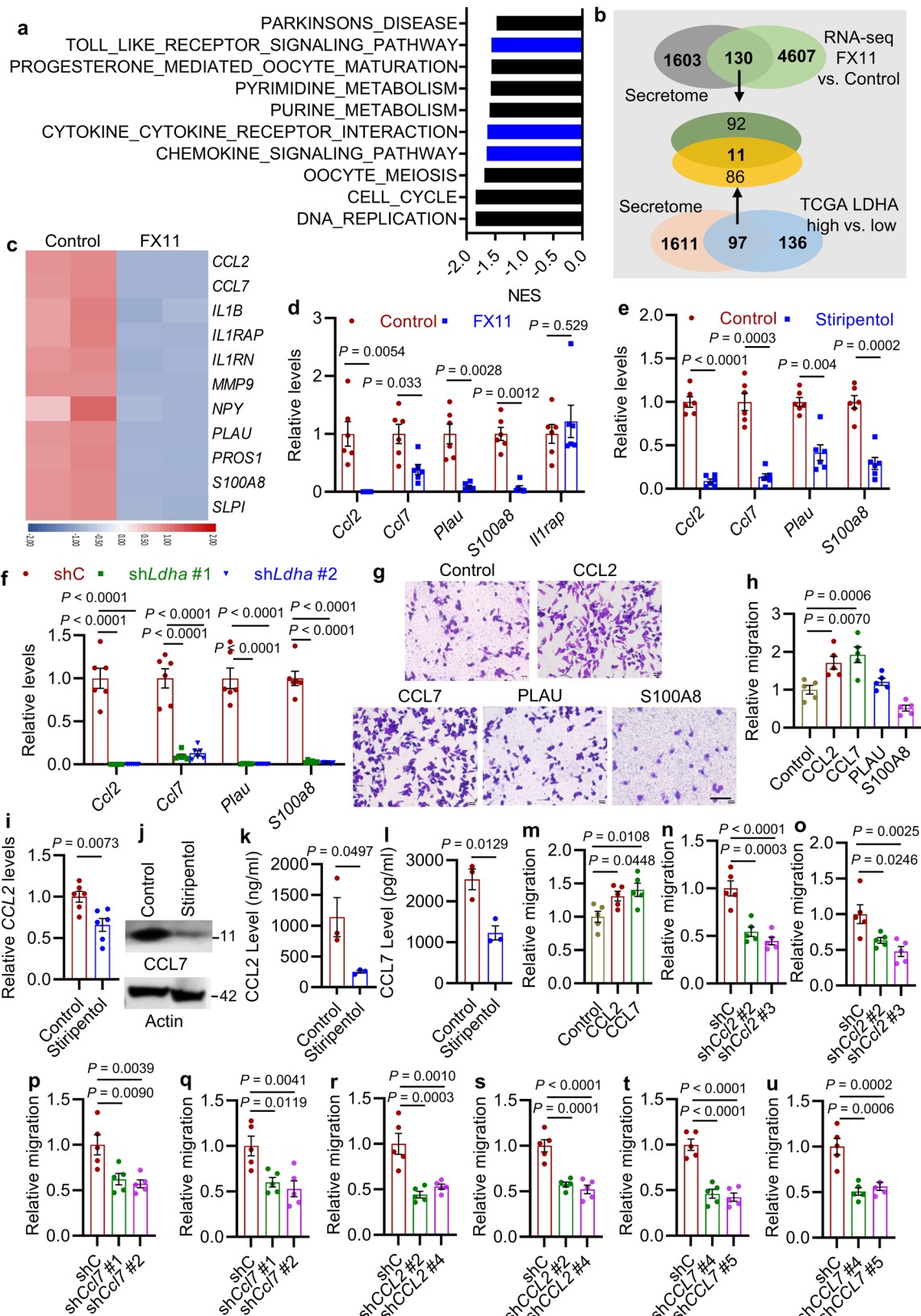

upregulation in both CT2A and GL261 cells (Fig. 5c, d). Next, we purified EVs from CM of EMφ using ultracentrifugation and confirmed their identity through performing the nanoparticle tracking analysis (Supplementary Fig. S10a) and Western blotting for EV markers (e.g., CD63 and ALIX) and calnexin that is absent from EVs (Fig. 5e). Notably, CT2A and GL261 CM treatment did not affect the size and distribution

of macrophage-derived EVs (Supplementary Fig. S10a), but increased LDHA levels in EMφ EVs, an effect that was abolished by shRNA-mediated *Ldha* depletion in macrophages (Fig. 5e). Moreover, EMφ EVs were labeled with the fluorescent dye 1,1′-dioctadecyl-3,3,3′,3′-tetramethylindocarbocyanine perchlorate (DiD) and then incubated with glioblastoma cells. Recipient glioblastoma cells exhibited equal

**Fig. 3 | LDHA promotes macrophage migration via upregulating CCL2 and CCL7. a** RNA-seq experiments and GSEA analysis in control and LDHA inhibitor FX11-treated CT2A cells. Top ten FX11-downregulated KEGG pathways are shown. Blue bars indicate the signatures relating to cytokine and chemokine pathways. **b** Identification of 11 genes encoding secreted proteins that are downregulated by FX11 treatment in CT2A cells and upregulated in *LDHA*-high glioblastoma patient tumors. **c** Heat map representation of the 11 downregulated genes in FX11-treated CT2A cells. Red and blue signal indicates high and low expression, respectively. **d** RT-qPCR for indicated genes in control and FX11-treated CT2A cells. The values were expressed as the fold change. *n* = 6 independent samples. **e, f** RT-qPCR for indicated genes in CT2A cells treated with or without stiripentol **e** or expressing shRNA control (shC) and *Ldha* shRNAs (sh*Ldha*) **f**. The values were expressed as the fold change. *n* = 6 independent samples. **g, h** Representative images (**g**) and quantification (**h**) of relative migration of Raw264.7 macrophages following stimulation with indicated recombinant proteins (10 ng/ml). Scale bar, 100 μm. *n* = 5 independent samples. **i** RT-qPCR for *CCL2* in GSC272 treated with or without stiripentol (10 μM). The values were expressed as the fold change. *n* = 6 independent samples. **j** Immunoblots of CCL7 in GSC272 treated with or without stiripentol (10 μM). **k, l** ELISA for CCL2 **k**, and CCL7 **l** in the conditioned media (CM) from number-matched GSC272 treated with or without stiripentol (10 μM). *n* = 3 independent samples. **m** Quantification of relative migration of human THP-1 macrophages following stimulation with recombinant CCL2 and CCL7 proteins (10 ng/ml). *n* = 5 independent samples. **n, o** Quantification of relative migration of Raw264.7 macrophages following stimulation with CM from CT2A (**n**) or GL261 (**o**) cells expressing shC and sh*Ccl2*. *n* = 5 independent samples. **p, q** Quantification of relative migration of Raw264.7 macrophages following stimulation with CM from CT2A (**p**) or GL261 (**q**) cells expressing shC and sh*Ccl7*. *n* = 5 independent samples. **r, s** Quantification of relative migration of THP-1 macrophages **r** and primary human BMDMs **s** following stimulation with CM from shC and sh*CCL2* GSC272. *n* = 5 independent samples. **t, u** Quantification of relative migration of THP-1 macrophages (**t**) and primary human BMDMs (**u**) following stimulation with CM from shC and sh*CCL7* GSC272 expressing. *n* = 5 independent samples. The experiments for (**d**–**j** and **m**–**u**) were independently repeated at least two times. Data presented as mean ± SEM and analysed by two-tailed Student's t-test (**d, e, i, k, l**) and one-way ANOVA test (**f, h, m**–**o, p**–**u**). Source data are provided as a Source Data file.

uptake efficiency for EVs from control macrophages, as well as CT2A EMφ and GL261 EMφ expressing shC and sh*Ldha* (Supplementary Fig. S10b–e). However, LDHA in glioblastoma cells was upregulated upon uptake of EVs from shC EMφ, but not from LDHA-depleted EMφ (Fig. 5f, g and Supplementary Fig. S10f, g). These findings reinforce the role of EVs in delivery of LDHA from EMφ to glioblastoma cells.

Next, we aimed to investigate the role of EMφ-derived EVs in regulating glioblastoma cell growth and survival. Among a series of cellular analyses in EMφ EV-treated glioblastoma cells, we found that EMφ EV treatment abolished *Ldha* knockdown-induced cell cycle transition from G1 to G2/M and apoptosis in CT2A (Supplementary Fig. S11a–d) and GL261 (Supplementary Fig. S11e–h) cells. Similarly, LDHA inhibitor stiripentol treatment-induced cell cycle transition from G1 to G2/M and apoptosis in CT2A and GL261 cells were rescued by the treatment with EVs from EMφ (Fig. 5h, i and Supplementary Fig. S12a–f). However, such effects were abolished or inhibited by depletion of LDHA in macrophages (Fig. 5h, i and Supplementary Fig. S12a–f).

Given the importance of glycolysis in tumor-macrophage symbiosis in glioblastoma, we further investigated the potential role of EMφ EVs in this process. Notably, we found that the impaired glycolytic activity as shown by reduced ECAR in LDHA-depleted glioblastoma cells was negated by the treatment with EVs from EMφ, but not LDHA-depleted EMφ (Fig. 5j). Moreover, treatment with EMφ EVs upregulated the levels of P-ERK, YAP1, P-STAT3, CCL2, and CCL7 in glioblastoma cells (Supplementary Fig. S12g, h). The decreased P-ERK, YAP1, P-STAT3, CCL2, and CCL7 in LDHA-depleted glioblastoma cells was rescued by the treatment with EMφ EVs, but not LDHA-depleted EMφ EVs (Fig. 5k–m). Together, these results demonstrate that TAM-derived LDHA-containing EVs can promote tumor growth by triggering a positive feedback loop between glioblastoma cell glycolysis and macrophage infiltration.

## Inhibition of LDHA-regulated tumor-macrophage symbiosis extends survival in glioblastoma mouse models

To further investigate the role of LDHA-mediated tumor-macrophage interplay in glioblastoma tumor biology, we utilized shRNA knockdown system to deplete LDHA in CT2A and GL261 tumors implanted into C57BL/6 mice. We found that LDHA depletion significantly inhibited tumor growth and extended survival in both glioblastoma mouse models (Fig. 6a, b and Supplementary Fig. S13a, b). Given the brain-penetrating ability of LDHA inhibitor stiripentol and isosafrole[42], we developed preclinical trials evaluating the anti-tumor effect of pharmacological inhibition of LDHA in glioblastoma mouse models. We found that stiripentol and isosafrole treatment impaired tumor growth and extended the survival of C57BL/6 mice implanted

with CT2A cells, GL261 cells, and 005 GSCs (Fig. 6c–e and Supplementary Fig. S13c–e). Moreover, we developed a patient-derived xenograft (PDX) model in nude mice by intracranial implantation of GSC272 and found that stiripentol treatment also extended survival (Fig. 6f). To confirm that macrophages were the critical target of stiripentol in impairing tumor growth and progression, we compared the anti-tumor effect of stiripentol and BLZ945 (an CSF-1R inhibitor that can impair macrophage role in mice) in GL261-bearing mice. Each agent extended survival; however, their combination treatment did not exhibit additional anti-tumor effects (Supplementary Fig. S13f). On the histological level, immunofluorescence for Ki67 and cleaved caspase 3 (CC3) demonstrated that glioblastoma cell proliferation was dramatically reduced, whereas apoptosis was increased upon *Ldha* depletion (Supplementary Fig. S14a–d) and treatment with stiripentol and isosafrole (Supplementary Fig. S14e–h). Flow cytometry demonstrated that macrophages were profoundly reduced in LDHA-depleted CT2A tumors (Supplementary Fig. S14i–k) and LDHA inhibitor-treated GL261 (Fig. 6g, h), CT2A (Fig. 6i and Supplementary Fig. S14l–n) and 005 GSC tumors (Supplementary Fig. S14o, p). Similarly, immunofluorescence for F4/80 confirmed that infiltrating macrophages were profoundly reduced in CT2A tumors by inhibition of LDHA genetically (Supplementary Fig. S14q, r) and pharmacologically (Supplementary Fig. S14s, t). However, LDHA inhibition with stiripentol did not change macrophage apoptosis in CT2A tumors (Supplementary Fig. S14u, v).

To confirm the role of LDHA in regulation of the YAP1/STAT3–CCL2/CCL7 signaling axis in vivo, we performed immunofluorescence for YAP1 and STAT3 in tumors and ELISA for CCL2 and CCL7 in plasma from control and LDHA-inhibited glioblastoma tumor-bearing mice. We found that the nuclear level of YAP1 and STAT3 in CT2A tumors (Fig. 6j-m) and plasma level of CCL2 and CCL7 from glioblastoma tumor-bearing mice (Fig. 6n, o) were significantly reduced upon stiripentol treatment. Similarly, blockade of the YAP1/STAT3 signaling using STAT3 inhibitor WP1066 reduced plasma level of CCL2 and CCL7 and intratumoral macrophages in CT2A-bearing mice (Fig. 6p–r and Supplementary Fig. S14w). Next, we investigated the in vivo role of CCL2 and CCL7 by implantation of shC, sh*Ccl2*, and sh*Ccl7* CT2A cells into the brains of C57BL/6 mice and found that depletion of CCL2 and CCL7 significantly extended survival (Fig. 6s) and reduced intratumoral macrophages (Fig. 6t and Supplementary Fig. S14x).

TAMs consist of pro-tumor and anti-tumor phenotypes and are usually biased toward a pro-tumor phenotype in glioblastoma[5,6,8,43]. We found that *LDHA* expression correlated positively with pro-tumor macrophage signature[32] in TCGA glioblastoma patient tumors (Supplementary Fig. S15a). CM from LDHA-inhibited (genetically and

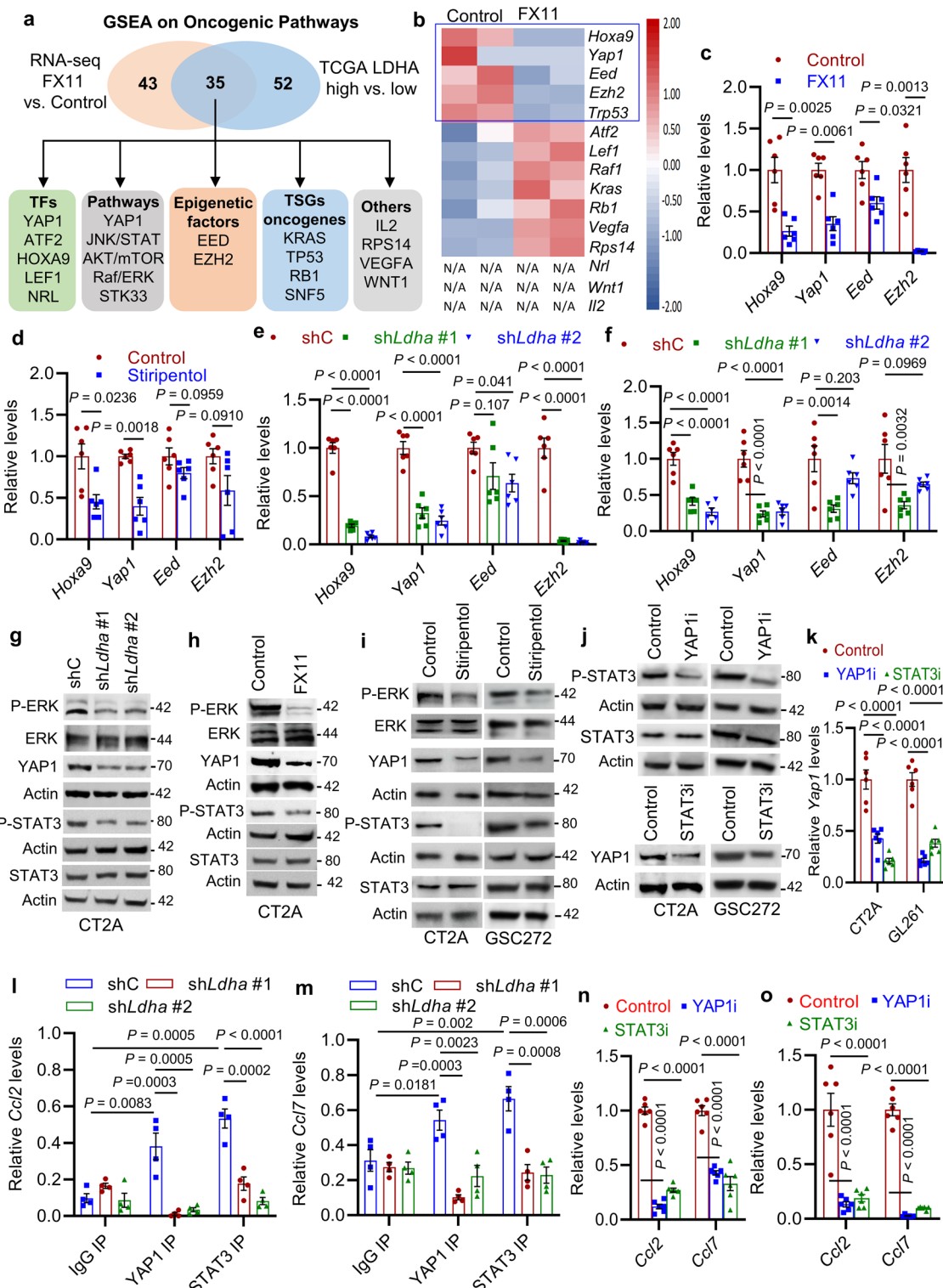

pharmacologically) CT2A and GL261 cells impaired the expression of a pro-tumor macrophage marker arginase 1 (Arg1) and the percentage of pro-tumor CD68+CD206+ cells in Raw264.7 macrophages (Supplementary Fig. S15b–g). Moreover, depletion of LDHA in glioblastoma cells or treatment with LDHA inhibitor stiripentol reduced pro-tumor CD45high CD11b+CD68+CD206+ macrophages in tumors from CT2A-bearing mice (Supplementary Fig. S15h-k). Similarly, pharmacologic inhibition of STAT3 or genetic depletion of CCL2 and CCL7 reduced pro-tumor CD45high CD11b+CD68+CD206+ macrophages in CT2A tumors (Supplementary Fig. S15l–o).

Finally, we aimed to investigate the role of TAM-derived LDHA-containing EVs in glioblastoma progression and treated sh*Ldha* CT2A-bearing mice with stiripentol and EMφ EVs. As expected, stiripentol treatment did not exhibit additional anti-tumor effects in LDHA-depleted tumors (Fig. 6u), supporting our above in vivo findings that LDHA is the key target of stiripentol. However, EMφ EVs treatment rescued the impaired tumor growth in LDHA-depleted CT2A tumors (Fig. 6u). To confirm the role of macrophage LDHA in this process, we generated macrophage-specific LDHA null (LDHA-mKO) mice by crossing LDHA flox mice with Lysozyme-Cre (LyzCre) mice. Orthotopic

**Fig. 4 | LDHA-induced CCL2 and CCL7 expression is regulated by YAP1 and STAT3 transcriptional co-activators. a** Identification of oncogenic pathways (using GSEA), including transcription factors (TFs), signaling pathways, epigenetic factors, tumor suppressor genes (TSG), oncogenes, and others that are downregulated by FX11 treatment in CT2A cells and enriched in *LDHA*-high glioblastoma patient tumors. **b** Heat map representation of above-identified factors in control and FX11-treated CT2A cells. The red signal indicates higher expression and the blue signal denotes lower expression. N/A indicates the gene that does not present in this dataset. The downregulated genes upon FX11 treatment are highlighted. **c, d** RT-qPCR for *Hoxa9*, *Yap1*, *Eed*, and *Ezh2* in CT2A cells treated with or without FX11 **c** or stiripentol **d**. The values were expressed as the fold change. *n* = 6 independent samples. **e, f** RT-qPCR for indicated genes in CT2A **e** and GL261 **f** cells expressing shRNA control (shC) and *Ldha* shRNAs (sh*Ldha*). The values were expressed as the fold change. *n* = 6 independent samples. **g, h** Immunoblots of P-ERK, ERK, YAP1, P-STAT3, and STAT3 in CT2A cells expressing shC and sh*Ldha* (**g**) or treated with or without FX11 (8 μM) (**h**). **i** Immunoblots of P-ERK, ERK, YAP1, P-STAT3, and STAT3 in CT2A cells and GSC272 treated with or without stiripentol (10 μM). **j** Immunoblots of P-STAT3 and STAT3 in CT2A cells and GSC272 treated with or without YAP-TEAD interaction inhibitor (YAP1i) verteporfin (1 μM); or immunoblots of YAP1 in CT2A cells and GSC272 treated with or without STAT3 inhibitor (STAT3i) WP1066 (10 μM). **k** RT-qPCR for *Yap1* in CT2A and GL261 cells treated with or without YAP1i (1 μM) or STAT3i (10 μM). The values were expressed as the fold change. *n* = 6 independent samples. **l, m** Quantification of YAP1 and STAT3 ChIP-PCR in the *Ccl2* **l** or Ccl7 **m** promoter of CT2A cells expressing shC and sh*Ldha*. *n* = 4 independent samples. **n, o** RT-qPCR for *Ccl2* and *Ccl7* in CT2A (**n**) and GL261 (**o**) cells treated with or without YAP1i or STAT3i. The values were expressed as the fold change. *n* = 6 independent samples. The experiments for (**c**–**m**) and (**n** and **o**) were independently repeated at least three and two times, respectively. Data presented as mean ± SEM and analysed by two-tailed Student's t-test (**c, d**) and one-way ANOVA test (**e, f, k, l, m, n, o**). Source data are provided as a Source Data file.

transplantation of CT2A cells into the brains of LDHA-mKO and WT mice showed significant survival extension in LDHA-mKO mice compared to WT mice (Fig. 6v). However, stiripentol treatment showed similar anti-tumor effects in both WT and LDHA-mKO mice (Fig. 6v). Together, these results validate the importance of LDHA-regulated tumor-macrophage symbiosis in promoting glioblastoma progression and support a therapeutic potential of targeting this co-dependency in glioblastoma.

### The LDHA–YAP1/STAT3–CCL2/CCL7 axis tracks with macrophages in glioblastoma patient tumors and is increased in glioblastoma patient plasma and EVs

The clinical relevance of above experimental findings was supported by bioinformatics using scRNA-seq data from 16 glioblastoma patients[34] showing that glioblastoma cell *LDHA*, *YAP1*, *STAT3*, and *CCL2* correlated positively with macrophage abundance (Fig. 7a). Moreover, bioinformatics analyses in TCGA glioblastoma dataset confirmed that *LDHA*, *YAP1*, *STAT3*, *CCL2*, and *CCL7* positively correlated with each other and with macrophage signature in patient tumors (Fig. 7b). Next, we performed immunofluorescence for LDHA and Mac-2 (a macrophage marker) in tumors from a cohort of 30 glioblastoma patients and found that LDHA signaling showed a positive correlation with the density of intratumoral macrophages (Fig. 7c, d). Since *LDHA*, *CCL2* and *CCL7* are genes encoding secreted proteins, we compared their protein levels in patient plasma showing that all of them were higher in glioblastoma patients than that in healthy controls, but such levels were not changed in meningioma patients (Fig. 7e–g). Moreover, plasma LDHA correlated positively with plasma CCL2 and CCL7 (Fig. 7h, i) and intratumoral macrophages in glioblastoma (Fig. 7j). Moreover, the median survival time of glioblastoma patients with high plasma LDHA (389 days) was lower than the patients with low plasma LDHA (675 days, Fig. 7k). However, it should be noted that this survival analysis did not reach statistical significance due to limited patient numbers (Fig. 7k). Moreover, plasma LDHA, CCL2, and CCL7 levels were not related to the status of recurrence, gender, age, and MGMT methylation in glioblastoma patients (Supplementary Fig. S16a–l). Finally, we isolated EVs from the plasma of a cohort of healthy controls and glioblastoma patients and confirmed their identity using electron microscopy (Fig. 7l) and nanoparticle tracking analysis (Supplementary Fig. S16m). The results from flow cytometry on these isolated EVs demonstrated that LDHA in glioblastoma patient plasma EVs was significantly higher than that from healthy controls (Fig. 7m, n and Supplementary Fig. S16n). Together, these correlative glioblastoma patient's findings are consistent with the hypothesis that LDHA–YAP1/STAT3–CCL2/CCL7 axis drives macrophage infiltration, and suggest that LDHA, CCL2 and CCL7 might function as biomarkers for glioblastoma patients, although these data are still relatively preliminary.

## Discussion

Glioblastoma cells can reprogram metabolic pathways to maintain their tumor potential. Aerobic glycolysis is used in tumors across cancer types (including glioblastoma) and is considered a hallmark of cancer[15]. However, whether and how aerobic glycolysis affects the biology of immune cells, such as macrophages, and, in turn, modulates tumor immunity and progression are not determined in glioblastoma. In this study, we screened a panel of metabolic small-molecule compounds and demonstrated that glioblastoma cell glycolysis is essential for macrophage infiltration. Mechanistically, LDHA-lactate-directed ERK pathway activates YAP1 and STAT3 transcriptional co-activators to upregulate CCL2 and CCL7 in glioblastoma cells, which promote macrophage infiltration into the TME. In addition to functioning as immunosuppressive cells inhibiting anti-tumor immunity[6,44], TAMs are known to promote glioblastoma cell proliferation and survival[3,6,8]. We provided further evidence showing that these infiltrating macrophages promote glioblastoma cell glycolysis, proliferation, survival, and tumor growth through the secretion of LDHA-containing EVs. Clinical validations demonstrated that the intratumoral LDHA–YAP1/STAT3–CCL2/CCL7 signaling axis and plasma LDHA track with macrophage density and may function as potential biomarkers for glioblastoma patients. Therefore, our current work reveals the molecular mechanisms underlying tumor-macrophage symbiosis and supports the hypothesis that targeting this LDHA-mediated symbiosis could provide clinical benefits for glioblastoma patients (Fig. 8).

Emerging evidence has shown that tumor-macrophage symbiotic interactions are critical for tumor progression[6,8,45]. Cancer cell metabolism not only provides sufficient energy for maintaining tumor growth but also affects the biology of myeloid cells (e.g., macrophages[19–21]) across cancer types, including glioblastoma[46,47]. LDHA is an aerobic glycolysis-related key enzyme contributing to lactate production in cancer cells[24,25]. Upon secretion, lactate plays an important role in regulating macrophage immunosuppressive polarization across cancer types, including breast cancer[48,49], lung cancer[50–52], melanoma[52], cervical cancer[53], and colon cancer[52]. In our study, we established that LDHA-mediated glioblastoma cell glycolysis promotes the infiltration of macrophages into the TME, which, in turn, supports tumor progression in glioblastoma mouse models. These results are consistent with the findings observed in multiple sclerosis, where enhanced glycolytic metabolism triggers the infiltration of macrophages[54]. Together, our work reinforces the importance of glioblastoma cell glycolysis in modulating the TME, particularly the infiltration of macrophages.

In exploring the connection between LDHA and macrophage biology, we demonstrated that glioblastoma cell LDHA upregulates multiple downstream chemokines, most prominently CCL2 and CCL7, to trigger macrophage infiltration, consistent with previous work[44,55]. Mechanistically, our study demonstrated that these two chemokines

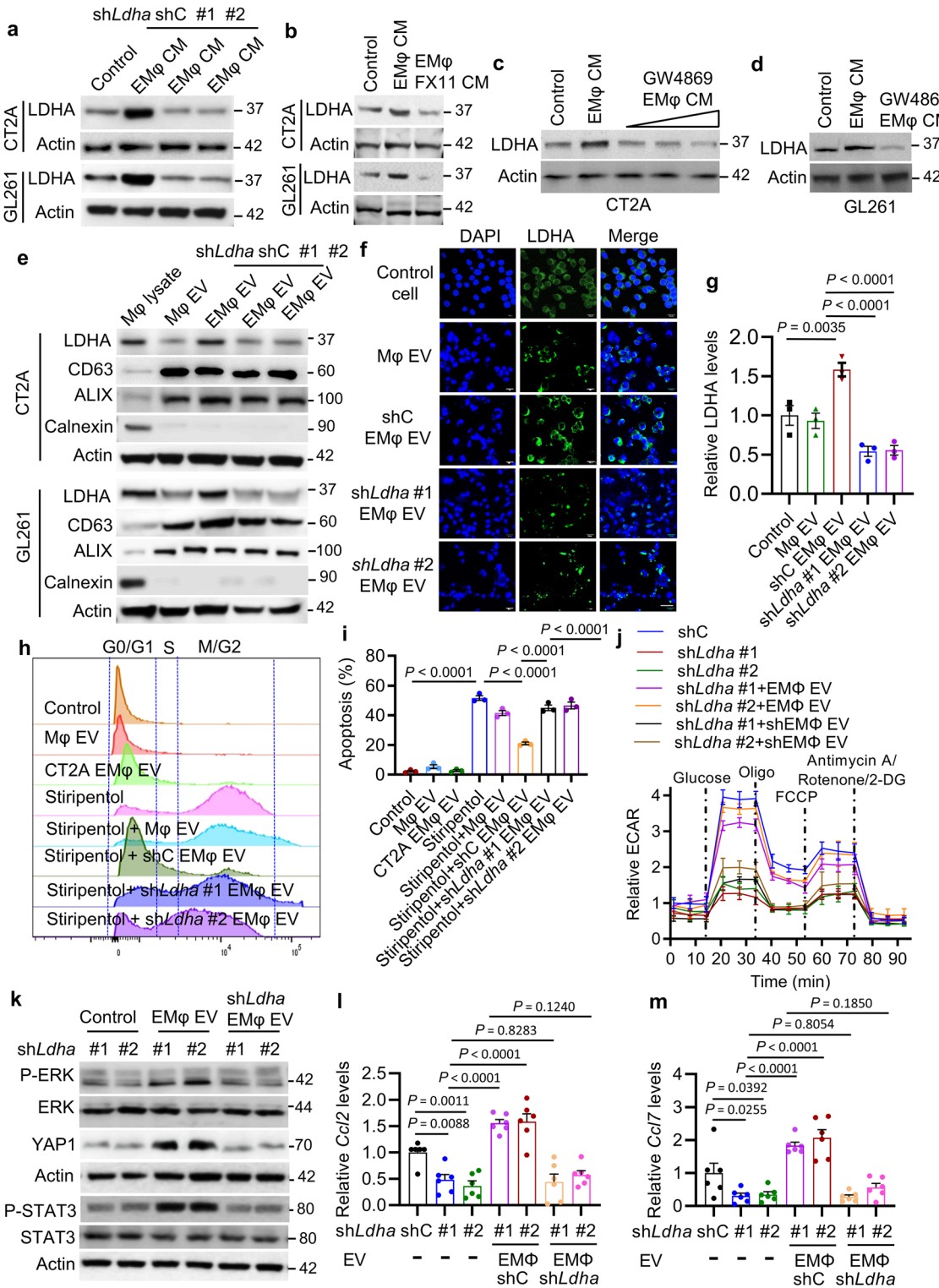

are regulated by LDHA/lactate-induced activation of YAP1 and STAT3 in glioblastoma cells. Moreover, we discovered that the ERK pathway is required for LDHA-induced YAP1 and STAT3 activation, which is consistent with previous work showing that ERK is downstream of LDHA in the heart[56] and breast cancer cells[57]. YAP1 is a transcription coregulator that plays a vital role in tumor progression[58]. In the context of glioblastoma, we have shown that YAP1 is essential for *PTEN* deficiency-induced transcriptional upregulation of LOX, which, in turn, triggers macrophage infiltration into the TME[3]. Here, we further identified that YAP1 activation promotes macrophage infiltration via direct

transcriptional regulation of CCL2 and CCL7 in glioblastoma cells, consistent with the findings observed in liver cancer[59,60]. These distinct YAP1-driven mechanisms underlying macrophage recruitment highlight a context-dependent tumor-macrophage symbiosis and the need for developing personalized medicine to target this symbiosis. STAT3 is a transcription factor that plays a critical role in regulating macrophage immunosuppressive polarization[61,62]. It is interesting to highlight the previous work showing that STAT3 transcriptionally upregulates LDHA in thyroid and bladder cancer cells[63,64]. Together with our findings, these work supports a reciprocal regulatory mechanism between LDHA

**Fig. 5 | TAM-derived LDHA-containing EVs promote glioblastoma cell growth and glycolysis, and activate the ERK-YAP1/STAT3-CCL2/CCL7 signaling.** **a** Immunoblots of LDHA in CT2A and GL261 cells treated with conditioned media (CM) from CT2A/GL261 CM-educated Raw264.7 macrophages (EMφ) expressing shRNA control (shC) or *Ldha* shRNAs (sh*Ldha*). **b** Immunoblots of LDHA in CT2A and GL261 cells treated with CM from EMφ in the presence or absence of FX11 (10 μM). **c** Immunoblots of LDHA in CT2A cells (control) and treated with CM from CT2A EMφ in the presence or absence of GW4869 at 1, 5, and 10 μM. **d** Immunoblots of LDHA in GL261 cells treated with or without CM from GL261 EMφ in the presence or absence of GW4869 at 10 μM. **e** Immunoblots of LDHA, CD63, ALIX, and calnexin in Raw264.7 macrophage lysate and in EVs isolated from Raw264.7 Mφ, CT2A EMφ and GL261 EMφ expressing shC and sh*Ldha*. **f, g** Representative images (**f**) and quantification (**g**) of immunofluorescence for LDHA in CT2A cells incubated with (500 ng) isolated from Raw264.7 Mφ and CT2A EMφ expressing shC and sh*Ldha* for 24 hrs. Scale bar, 200 μm. *n* = 3 independent samples. **h** Representative images of cell cycle analysis of CT2A cells treated with EVs (500 ng) isolated from Raw264.7

Mφ and CT2A EMφ, as well as with stiripentol (10 μM) in the presence or absence of EVs isolated from CT2A EMφ expressing shC and sh*Ldha*. A representative example of three replicates. **i** Quantification of flow cytometry apoptosis analysis in CT2A cells treated with EVs (500 ng) isolated from Raw264.7 Mφ and CT2A EMφ, as well as with stiripentol (10 μM) in the presence or absence of EVs isolated from CT2A EMφ expressing shC and sh*Ldha*. *n* = 3 independent samples. **j** Extracellular acidification rate (ECAR) of CT2A cells expressing shC and sh*Ldha* and treated with or without EVs (500 ng) isolated from CT2A EMφ and sh*Ldha* EMφ. *n* = 6 independent samples. **k** Immunoblots of P-ERK, ERK, YAP1, P-STAT3, STAT3, and Actin in LDHA-depleted CT2A cells treated with or without EVs (500 ng) isolated from CT2A EMφ and sh*Ldha* EMφ. **l, m** RT-qPCR for *Ccl2* **l** and *Ccl7* **m** in LDHA-depleted CT2A cells treated with or without EVs (500 ng) isolated from CT2A EMφ and sh*Ldha* EMφ. *n* = 6 independent samples. The experiments for (**a–m**) were independently repeated at least three times. Data presented as mean ± SEM and analysed by one-way ANOVA test (**g, i, l, m**). Source data are provided as a Source Data file.

and STAT3, which may induce a potent feedback loop to promote macrophage infiltration through CCL2 and CCL7 production. Our results of CCL2 and CCL7 as downstream signals of STAT3 are consistent with previous work focusing on fibroblasts in breast cancer[65] and on muscle satellite cells in injured muscles[66]. Consistent with previous report[40], our work highlights that STAT3 and YAP1 are transcriptional co-activators that coordinately upregulate CCL2 and CCL7 in glioblastoma cells, thus stimulating macrophage infiltration into the TME.

Macrophages are the most prominent immune cells in the glioblastoma TME. As a result of infiltration, they promote tumor growth and progression by secreting distinct soluble factors, including various growth factors, cytokines, and EVs[8,22]. EVs can transfer proteins, RNA, microRNAs, DNA, and metabolites from parent cells to recipient cells, thus promoting tumor progression[67]. In our study, analysis of scRNA-seq data from glioblastoma patient tumors demonstrated that LDHA is highly expressed by both glioblastoma cells and macrophages. Functional studies demonstrated that EMφ CM treatment upregulates LDHA levels in glioblastoma cells, and this effect is abolished when EMφ were pretreated with EV biogenesis inhibitor, LDHA inhibitor, or harboring LDHA knockdown/KO, suggesting that LDHA can be transferred from EMφ to glioblastoma cells. In addition to supporting previous studies focusing on a cell-autonomous role of LDHA in cancer cells[68], including glioblastoma cells[69,70], our work reinforces the view that LDHA is a key molecule controlling the symbiotic interactions between glioblastoma cells and macrophages, and highlights the critical role of this symbiosis in promoting glioblastoma cell proliferation and survival.

After dissecting the molecular mechanisms underlying tumor-macrophage symbiosis, we investigated the biological and clinical impact of targeting this symbiosis in glioblastoma. We have shown that genetic depletion of LDHA in glioblastoma cells or macrophages extends survival, reduces macrophage infiltration and glioblastoma cell proliferation, and promotes glioblastoma cell survival in mouse models. In line with these findings from mouse models, analysis of tumor and plasma samples from glioblastoma patients demonstrates that the LDHA−YAP1/STAT3−CCL2/CCL7 signaling axis tracks with macrophages. Together, the identification of tumor-macrophage symbiosis, coupled with the anti-tumor effect of LDHA inhibition in glioblastoma mouse models and clinical validations, encourages the development of therapeutic strategies targeting this symbiosis in glioblastoma patients. Emerging evidence highlights that pharmacological targeting of tumor-macrophage symbiosis is a promising strategy for glioblastoma treatment, and multiple approaches, including CSF-1R inhibition, have been proposed[5]. Previous studies have shown that CSF-1R inhibitors can impair tumor progression and decrease immunosuppressive macrophages in glioblastoma mouse models[71–73]. However, these treatments result in therapy resistance due to enhanced PI3K activity in glioblastoma cells driven by macrophage-derived insulin-like growth factor-1 (IGF-1)[73]. Correspondingly, clinical trials with CSF-1R inhibition failed in

patients with glioblastoma[74] and resulted in serious side effects since CSF-1R is also expressed on monocytes and other stromal cells[75]. In this study, we developed preclinical trials in glioblastoma mouse and PDX models with LDHA inhibitors stiripentol and isosafrole[42] and found that these treatments extend the survival of tumor-bearing mice via blockade of tumor-macrophage symbiosis. Stiripentol is an FDA-approved antiepileptic drug for Dravet syndrome, a severe genetic brain disorder[76,77]. Isosafrole is a stiripentol analog that significantly inhibits the pyruvate-to-lactate conversion and suppresses seizures in a mouse model of epilepsy[42]. Based on the nature (e.g., well-tolerated in patients and BBB penetrating ability) of the two compounds, coupled with their anti-tumor effect in glioblastoma mouse and PDX models, we anticipate a tremendous translational potential of LDHA inhibition to improve patient outcomes.

In summary, our work reveals that glioblastoma cell glycolysis triggers the infiltration of macrophages into the TME via upregulating LDHA-regulated CCL2/CCL7, and reciprocally, macrophages promote tumor growth and survival via EVs delivering LDHA to glioblastoma cells. Therefore, targeting LDHA-mediated tumor-macrophage symbiosis using the BBB penetrable compounds (e.g., stiripentol and isosafrole) is a promising strategy for treating patients with glioblastoma.

## Methods
### Mice and intracranial xenograft tumor models
All animal experiments were performed with the approval of the Institutional Animal Care and Use Committee at Northwestern University (protocol numbers IS00017931, IS00016006, and IS00015772). Female C57BL/6 (Jackson Laboratory, #0000664) and nude (Jackson Laboratory, #007850) mice at 5-6 weeks of age were grouped by 5 animals and maintained under pathogen-free conditions. LDHA-flox mice (Jackson Laboratory, #030112) were crossed with LyzCre mice (Jackson Lab, #004781) to obtain LDHA-mKO mice. Animals were housed in temperature- (21–23 °C) and humidity- (30–70%) controlled rooms with 12:12 light/dark cycles. The intracranial xenograft tumor models in C57BL/6 and nude mice were established as we described previously[3]. In brief, mice were anesthetized by intraperitoneal injection of a stock solution containing ketamine (Covetrus, #056344, 100 mg/kg) and xylazine (Akorn, #59399-110-20, 20 mg/kg) and were placed into the stereotactic apparatus (RWD Life Science, # 68513). A small hole was bored in the skull 1.2 mm anterior and 3.0 mm lateral to the bregma using a dental drill. Cells were injected in a total volume of 5 μl into the right caudate nucleus 3 mm below the brain surface using a 10 μl Hamilton syringe with an unbeveled 30-gauge needle. The incision was closed using tissue adhesive (3 M Vetbond, #1469SB). Mice were treated with LDHA inhibitor stiripentol (MCE MedChemExpress, #HY-103392; 150 mg/kg, i.p.) and its analog isosafrole (Chem Service, #120-58-1; 150 mg/kg, i.p.), CSF-1R inhibitor BLZ945 (Selleck Chemicals, #S7725;

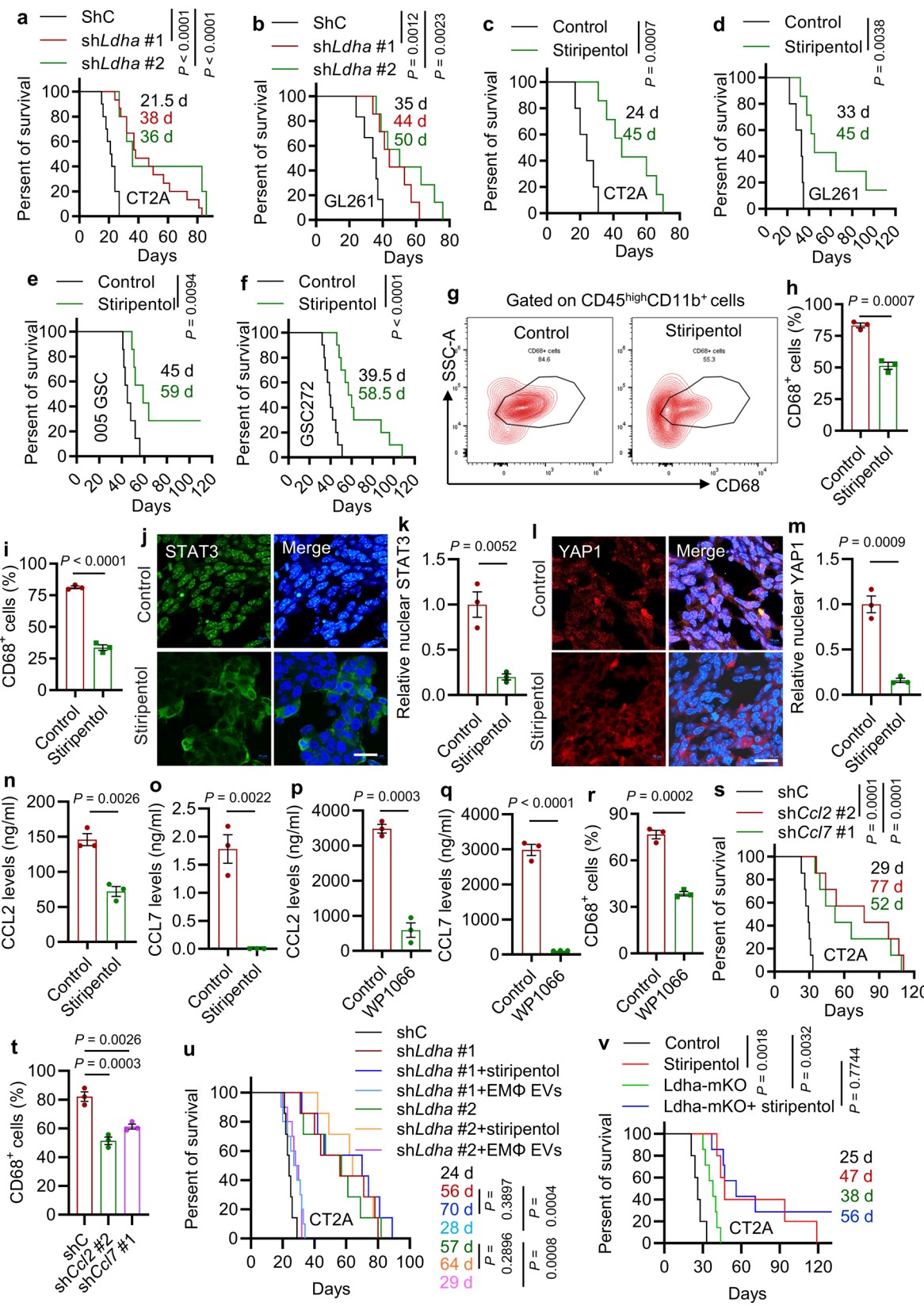

200 mg/kg, oral gavage), or STAT3 inhibitor WP1066 (Selleck Chemicals, # S2796; 60 mg/kg, oral gavage). Mice with neurologic deficits or moribund appearance were sacrificed. Following the transcardial perfusion with 4% PFA, brains were removed and fixed in formalin (Fisher Chemical, #SF100-4), and were processed for paraffin-embedded blocks or OCT-embedded blocks.

## Cell culture

CT2A, THP-1 macrophages, and 293 T cells were cultured in Dulbecco's Modified Eagle's Medium (DMEM; Gibco, #11995-065) containing 10% FBS (Fisher Scientific, # 16140071) and 1:100 antibiotic-antimycotic (Gibco, #15140-122), and were purchased from the American Type Culture Collection (ATCC). GL261 cells were cultured in DMEM-Ham's F12

**Fig. 6 | Inhibition of LDHA-mediated tumor-macrophage symbiosis reduces macrophage infiltration and glioblastoma growth in vivo. a, b** Survival curves of C57BL/6 mice implanted with $2 \times 10^4$ CT2A cells **a** expressing shRNA control (shC, $n = 10$ mice) and *Ldha* shRNAs ($n = 15$ and 5 mice for sh*Ldha* #1 and sh*Ldha* #2 group, respectively) or GL261 cells **b** expressing shC ($n = 6$ mice) and sh*Ldha* ($n = 7$ mice). **c–e** Survival curves of C57BL/6 mice implanted with CT2A **c** and GL261 cells **d**, or $1 \times 10^5$ 005 GSCs **e**. Mice were treated with stiripentol (150 mg/kg, i.p., every other day for 6 doses) beginning at day 8 **c, d**, $n = 5$ and 7 mice for control and stiripentol group, respectively) or 11 **e**, $n = 7$ mice per group) post-orthotopic injection. **f** Survival curves of nude mice implanted with $2 \times 10^5$ GSC272 and treated with stiripentol (150 mg/kg, i.p., every other day, 8 doses) beginning at day 15 post-orthotopic injection. $n = 10$ mice per group. **g–i** Representative **g** and quantification **h** of flow cytometry analysis for the percentage of CD68+ macrophages out of CD45high CD11b+ cells in GL261 **g, h** or CT2A **i** tumors treated with or without stiripentol. $n = 3$ independent samples. **j–m** Immunofluorescence and quantification of nuclear STAT3 **j, k** or YAP1 **l, m** positive cells in CT2A tumors treated with or without stiripentol. Scale bar, 20 μm. $n = 3$ independent samples. **n, o** The plasma level CCL2 and CCL7 in GL261 tumor-bearing mice treated with or without stiripentol **n, o** or in CT2A tumor-bearing mice treated with or without STAT6 inhibitor WP1066 (60 mg/kg, oral gavage, every other day for 6 doses; **p, q**. $n = 3$ independent samples.

**r** Quantification of flow cytometry analysis for the percentage of CD68+ macrophages out of CD45high CD11b+ cells in CT2A tumors treated with or without WP1066. $n = 3$ independent samples. **s** Survival curves of C57BL/6 mice implanted with CT2A cells expressing shC, sh*Ccl2* or sh*Ccl7*. $n = 10$ mice per group. **t** Quantification of flow cytometry analysis for the percentage of CD68+ macrophages out of CD45high CD11b+ cells in shC, sh*Ccl2* and sh*Ccl7* CT2A tumors. $n = 3$. **u** Survival curves of C57BL/6 mice implanted with CT2A cells expressing shC and sh*Ldha*. Mice were treated with or without stiripentol and extracellular vesicles (EVs, 5 μg/mouse, i.v., every other day for five doses) isolated from CT2A CM-treated Raw264.7 macrophages (EMΦ EVs) beginning at day 8 post-orthotopic injection. $n = 7$ mice per group except for the groups of sh*Ldha* #1 + MΦ EVs or sh*Ldha* #2 + MΦ EVs where $n = 10$ mice per group. **v** Survival curves of WT and LDHA-mKO mice implanted with CT2A cells and treated with or without stiripentol beginning at day 8 post-orthotopic injection. $n = 7$ mice per group except for the groups of control and stiripentol where $n = 5$ mice per group. The experiments for (**j–m**) were independently repeated at least three times. Data presented as mean ± SEM. Statistical analyses were determined by log-rank test (**a–f, s, u, v**), two-tailed Student's t-test (**h, i, k, m–r**), and one-way ANOVA test (**t**). Source data are provided as a Source Data file.

medium (Gibco, #10565-018) containing 10% FBS and 1:100 antibiotic-antimycotic. Raw264.7 macrophages were cultured in RPMI 1640 medium (RPMI, Gibco, #22400-089) containing 10% FBS and 1:100 antibiotic-antimycotic. These cell lines were purchased from ATCC. SB28 cell line was provided by Dr. Hideho Okada (UCSF), and cultured in RPMI 1640 supplemented with 10% FBS, 1% MEM-NEAA, 1% HEPES, 1% Sodium Pyruvate, 1% Glutamax, 0.1% β-mercaptoethanol, and 1:100 antibiotic-antimycotic. The mouse glioblastoma tumor-derived GSC lines 005 GSC and QPP7 were provided by Dr. S.D. Rabkin (Massachusetts General Hospital, Boston) and Dr. J. Hu (MD Anderson Cancer Center, Houston), respectively. Human GSC272 was provided by Dr. Frederick Lang (MD Anderson Cancer Center, Houston). GSCs were cultured in neural stem cell (NSC) proliferation media (Millipore, #SCM005) containing 20 ng/ml basic fibroblast growth factor (bFGF; PeproTech, #100-18B) and 20 ng/ml epidermal growth factor (EGF; PeproTech, #AF-100-15). All cells were maintained at 37 °C and 5% $CO_2$ and confirmed to be mycoplasma-free. Conditioned media (CM) were collected from number-matched shC and shRNA knockdown cells, or control and compound-pretreated (24 hrs) cells after culturing for another 24 hrs in FBS-free (growth factor-free for GSCs) and compound-free culture medium.

### Isolation and culture of primary BMDMs
Primary mouse BMDMs were isolated from C57BL/6 mice and cultured as we described previously[3,49]. For human BMDMs, we isolated CD34+ hematopoietic stem and progenitor cells from bone marrow aspirates of a female donor. Bone marrow cells were diluted with sterile PBS (1:1) without $Ca^{2+}$ and $Mg^{2+}$, and layered on top of an equal volume of Ficoll Paque Premium (Sigma Aldrich, #17-5442-02). Samples were then centrifuged at 300 ×g for 40 min at room temperature without brake, the plasma layer was removed, and the buffy coat containing mononuclear cells was extracted. Mononuclear cells were blocked using FcR block (Miltenyi, #130-059-901) and treated with CD34 microbeads (Miltenyi, #130-100-453) according to manufacturer's dilution instructions. Following incubation, cells were applied to positive selection columns (Miltenyi, #130-04-401) on a QuadroMACS Separator and finally eluted with sterile PBS. Cells were then differentiated in Serum-Free Expansion Medium (SFEM; Stem-Cell Technologies, #09650) containing 100 ng/ml stem cell factor (SCF; R&D Systems, #255-SC-050/CF), 50 μg/ml thrombopoietin (TPO; Peprotech, #300-18), 50 ng/ml FMS-like tyrosine kinase 3 ligands (FLT3L; R&D Systems, #308-FK-250/CF), 50 ng/ml macrophage colony-stimulating factor (M-CSF, Peprotech, #300-25), 20 ng/ml IL6 (Peprotech, #200-06), and 10 ng/ml IL3 (Peprotech, #200-06) for 14-21 days. Differentiation was signaled by the appearance of

adherent macrophages and confirmed by flow cytometry analysis using anti-CD11b (Biolegend, #301356) and anti-CD14 (Biolegend, #325608). Isolation and culture of human BMDMs were performed under the IRB protocol #P00031718 at Boston Children's Hospital.

### Plasmids and viral transfections
shRNAs targeting mouse *Ldha, Ccl2* and *Ccl7* in the pLKO.1 vector (Sigma, #SHC001) were used in this study. Lentiviral particles (8 μg) were generated by transfecting 293 T cells with the packaging vectors pMD2.G (2 μg; Addgene, #12259) and psPAX2 (4 μg; Addgene, #12260). Lentiviral particles were collected at 48 and 72 hrs after transfection into 293 T cells. Receiving cells were infected with viral supernatant containing 10 μg/mL polybrene (Millipore, #TR-1003-G). After 48 hrs, infected cells were selected using puromycin (2 μg/ml; Millipore, #540411) and assessed for the expression of LDHA, CCL2, and CCL7 by immunoblots or RT-qPCR. The following shRNA sequence: *Ldha*: #1: TRCN0000041743 and #2: TRCN0000041744; mouse *Ccl2*: #2: TRCN0000034472 and #3: TRCN0000034473; mouse *Ccl7*: #1: TRCN0000068135 and #2: TRCN0000068136), human CCL2: #2: TRCN0000006281 and #4: TRCN0000006283, and human CCL7: #4: TRCN0000057896 and #5: TRCN0000057897 were selected for further use following validation.

### Migration assay
Macrophages ($1 \times 10^4$ for Raw264.7 and BMDMs and $5 \times 10^5$ for THP-1 macrophages) were suspended in serum-free culture medium and seeded into 24-well Transwell inserts (5.0 μm, Corning, #07-200-149). Conditioned media (CM) from glioblastoma cells and GSCs or normal medium with indicated factors were added to the remaining receiver wells. After 8 hrs (Raw264.7 macrophages and mouse primary BMDMs) or 16 hrs (THP-1 macrophages and human primary BMDMs), the migrated macrophages were fixed and stained with crystal violet (0.05%, Sigma, #C-3886), and then cells per field of view were counted under the microscope. Moreover, we performed the scratch would healing assay on macrophages treated with or without CM from control and LDHA-depleted/inhibited glioblastoma cells using a protocol as we reported previously[78].

### Metabolic compound screen
For the initial screening, CT2A cells were seeded in 6-well plated and treated with 55 compounds with metabolic reprogramming function from the CNS-Penetrant Compound Library (MCE Med-ChemExpress, #HY-L028) at 10 μM for 24 hrs. After the treatment with the compounds for 24 hrs, CT2A cells were then cultured with

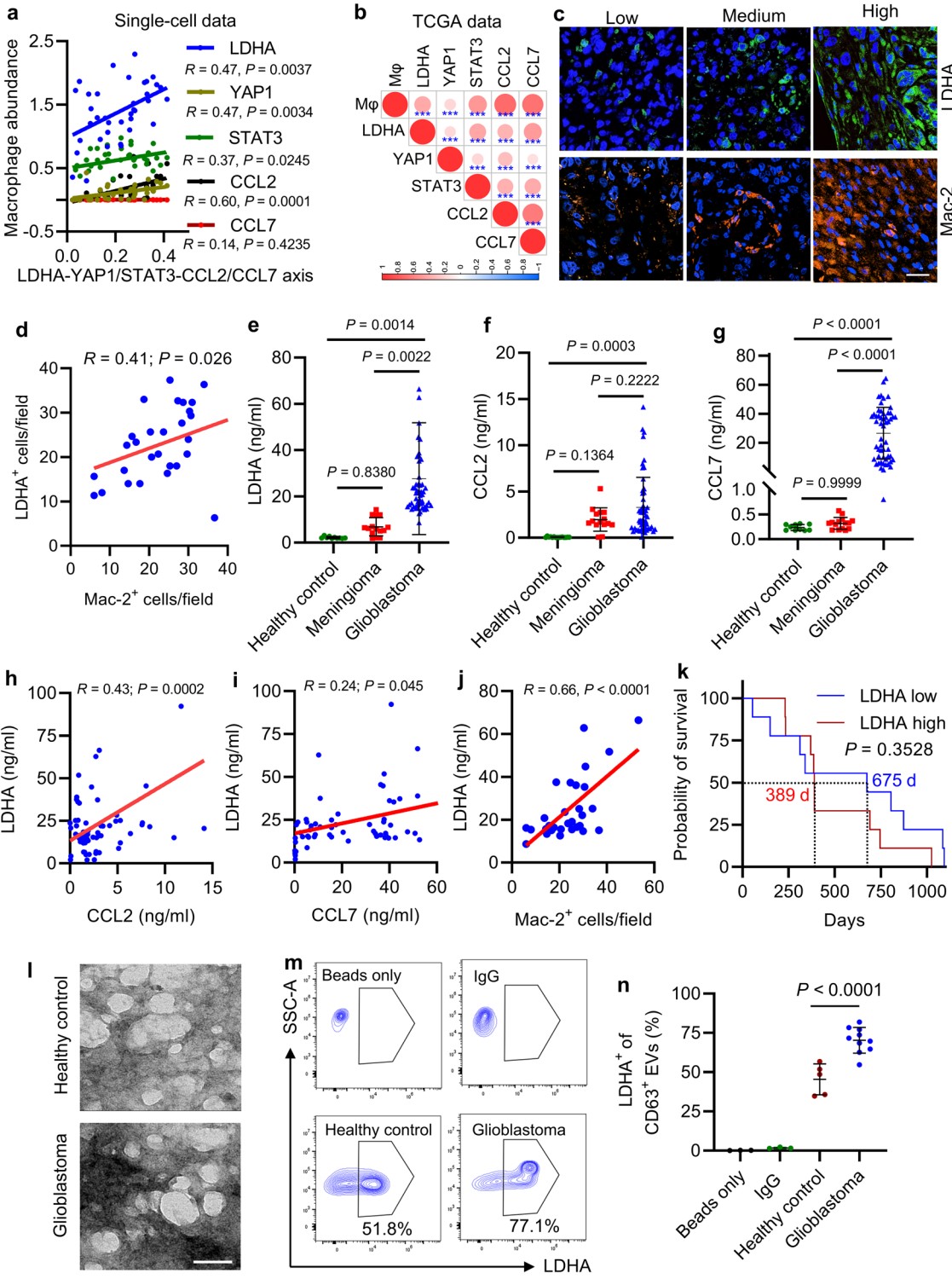

FBS- and compound-free culture medium for additional 24 hrs. The conditioned media (CM) from number-matched control and compound-treated CT2A cells were collected and used for Raw264.7 macrophage transwell migration assay. The compounds with a significant effect on inhibiting CT2A cell CM-induced macrophage migration were selected for a second round of screen at a lower concentration (5 μM).

## Colony formation assay
Colony formation assay was used to examine glioblastoma cell proliferation in vitro. In brief, 1500 glioblastoma cells were seeded and cultured for about 8 days in each well of 6-well plates. Finally, cells were fixed and stained with 0.5% crystal violet for 1 hr. These experiments were performed in triplicate.

## Cell cycle and apoptosis analysis
Cells were cultured in 6-well plates for 24 hrs, and fixed in ice-cold 70% ethyl alcohol for 30 min at 4 °C. For cell cycle analysis, cells were incubated with RNase A solution (Promega, #A797C; 100 μg/ml) for 5 min at room temperature and then stained with propidium iodide (PI) labeled with RedX (Biolegend, #421301, 50 μg/ml) for 10 min at 4 °C. PI incorporation was analyzed by flow cytometry. For apoptosis

**Fig. 7 | The LDHA–YAP1/STAT3–CCL2/CCL7 axis tracks with macrophages in glioblastoma patients and is increased in patient plasma EVs. a** The correlation of glioblastoma cell LDHA, YAP1, STAT3, CCL2, and CCL7 with macrophage abundance in glioblastoma patient tumors based on single-cell RNA sequencing data[34] ($n = 37$). R and P values are shown. **b** The correlation of LDHA, YAP1, STAT3, CCL2, and CCL7 with the abundance of macrophages and monocytes in glioblastoma patient tumors based on TCGA dataset ($n = 478$). Red signal indicates positive correlation and blue signal denotes negative correlation. ***$P < 0.0001$.
**c, d** Representative images **c** and correlation quantification analysis **d** between LDHA and Mac-2 expression in glioblastoma patient tumors ($n = 30$). Scale bar, 50 μm. R and P values are shown. **e–g** ELISA for LDHA **e**, CCL2 **f**, and CCL7 **g** in the plasma from healthy controls ($n = 10$), meningioma ($n = 15$), and glioblastoma ($n = 54$) patients. **h** Correlation analysis between plasma LDHA level and plasma CCL2 level in meningioma ($n = 15$), and glioblastoma ($n = 54$) patients. R and P values are shown. **i** Correlation analysis between plasma LDHA level and plasma

CCL7 level in meningioma ($n = 15$), and glioblastoma ($n = 54$) patients. R and P values are shown. **j** Correlation analysis between plasma LDHA level and intratumoral macrophage density (Mac-2⁺ cells) in glioblastoma patients ($n = 30$). R and P values are shown. **k** Kaplan-Meier survival curves of glioblastoma patients relative to high (top 25%, $n = 9$) and low (bottom 25%, $n = 9$) serum LDHA level. The median survival time of each group is indicated. Log-rank test. **l** Transmission electron microscopy analysis of extracellular vesicles (EVs) isolated from the plasma of healthy control and glioblastoma patients ($n = 5$). Scale bar, 100 nm.
**m, n** Representative images (**m**) and quantification (**n**) of flow cytometry for LDHA in CD63⁺ EVs isolated from the plasma of healthy controls ($n = 5$) and glioblastoma patients ($n = 10$). The experiments for (**l–n**) were independently repeated at least three times. Data presented as mean ± SD. Statistical analyses were determined by Pearson's correlation test (**a, b, d, h–j**), one-way ANOVA test (**e–g**), and two-tailed Student's t-test (**n**). Source data are provided as a Source Data file.

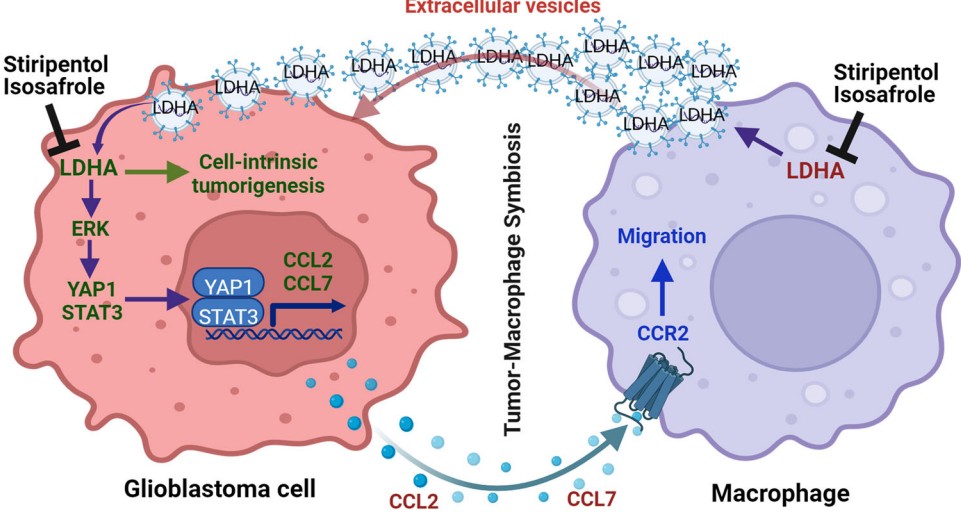

**Fig. 8 | Working model.** Schematic representation of the role of the LDHA–ERK–STAT3/YAP pathway in regulation of CCL2 and CCL7 in glioblastoma cells, which, in turn, promotes macrophage infiltration. These infiltrating macrophages are educated by the TME and promote glioblastoma cell proliferation and survival via transferring LDHA-containing extracellular vesicles. Inhibition of LDHA is a promising therapeutic strategy for glioblastoma via blockade of the tumor-macrophage symbiotic interaction. This image was created with BioRender.com.

analysis, cells were incubated with FITC-conjugated annexin V (BioLegend, #640906) and PI labeled with RedX (1 μg/ml) for 15 min at room temperature and analyzed using a flow cytometer.

## Metabolic assays
Lactate levels were measured using a glycolysis assay kit (Sigma-Aldrich, #MAK439) according to the instruction. Briefly, control and glycolysis/LDHA-inhibited cells were seeded in 96-well plate at a density of $3 \times 10^5$ cells and cultured in 1% FBS culture media containing with or without glucose (55 mM). Following the collection of CM, glycolytic activity (Lactate level) was measured at different time intervals for 1.5 hrs at 565 nm wavelength. On the other hand, glucose metabolism of indicated control and/or treated/modified glioblastoma cells and macrophages was measured using the Seahorse XF Cell Mito Stress Test Kit (Agilent Technologies, #103015-100) in a Seahorse XFe96 analyzer on Seahorse XFe96/XF Pro FluxPak Mini plates (Agilent Technologies, #103793-100) as instructed by the manufacturer's protocol.

## Enzyme-linked immunoassay
The levels of LDHA, CCL2, and CCL7 in human plasma or CM of GSC272, and CCL2 and CCL7 in plasma from GL261 tumor-bearing mice were measured by enzyme-linked immunoassay (ELISA) using the commercial human LDHA kit (Biomatik, #EKE60382), human CCL2 kit (Sigma-Aldrich, #RAB0054), human CCL7 kit (Sigma-Aldrich, #RAB0078),

mouse CCL2 kit (Sigma-Aldrich, #RAB0055), and mouse CCL7 kit (Invitrogen, #BMS6006INST) following the manufacturer's instructions.

## Extracellular vesicle isolation
For EV isolation from cells, indicated cells were grown with vesicle-depleted FBS for 24 hrs, and conditioned media were collected and centrifuged at 300 x g for 10 min, 2000 x g for 10 min, and 10000 x g for 30 min to remove cell debris. The supernatant was filtered through a 0.2 μm filter and centrifuged at 100,000 x g 4 °C for 70 min. The pellets were resuspended in cold-cold PBS and applied for the second round of ultracentrifugation. Finally, the pellets containing EVs were resuspended in 100 μl ice-cold PBS for further experiments. For EV isolation from human plasma, the SmartSEC Single EV Isolation System (System Biosciences, #SSEC200A-1) was used according to the manufacturer's instructions. Briefly, plasma samples with an additional column buffer of up to 4 ml were placed directly into the pre-washed column, incubated for 30 min at room temperature, and centrifuged at 500 xg for 1 min to elute the EVs in the flow through. The EV's supernatant was used for further flow cytometry analysis.

## Nanoparticle tracking analysis
Concentrated EVs were diluted using freshly filtered PBS and analyzed using a NanoSight NS3000 device (Nanosight, Malvern). A monochromatic laser beam at 405 nm was set to analyze the nanoparticles, and a video with a 30-second duration was taken at a rate of 30 frames

per second. Approximately 30–100 particles were analyzed in each field of view, and then particle brown-movement was assessed using the nanoparticle tracking analysis (NTA) software (version 2.3, Nanosight). NTA post-acquisition settings were optimized, and recorded video was analyzed to measure particle sizes and concentrations.

## Transmission electron microscopy

Isolated EVs were fixed with 2% paraformaldehyde (PFA; Alfa Aesar, #J61899), and deposited on pure carbon-coated transmission electron microscopy (TEM) grids for 20 min in a dry environment. The grids were washed with PBS 2 times and stained with 1.5% uranyl acetate for 5 min. After drying at room temperature, the grids were viewed under an FEI Tecnai G2 Spirit Twin TEM.

## EV internalization assay

EVs were labeled with DiD fluorescent dye (Biotium, #60014) for 30 min on a shaker. Then, the DiD-labeled EVs were added to tumor cell culture medium and incubated for 24 hrs at 37 °C. Cells were fixed with 4% PFA for 15 min and counterstained with 4′,6-diamidino-2-phenylindole (DAPI)/anti-fade mounting medium (Vector Laboratories, #H-1200-10) before confocal microscope (Nikon) examination.

## Flow cytometry

For intratumoral macrophage analysis, the tumor single-cell suspensions were incubated with fixable viability dye (Invitrogen, #5211229035) at room temperature for 10 min. After washing with FACS buffer (PBS with 1% BSA), cells were incubated with following antibodies: CD45 (BioLegend, #103132), CD11b (BioLegend, #101216), CD68 (BD Pharmingen, #566386), and CD206 (BD Bioscience, #565250) for 30 min at room temperature. After staining, cells were washed twice with FACS buffer and then fixed with 1% PFA/FACS buffer at 4 °C before performing flow cytometry analysis. For EV analsysis, CD63 exosome capture beads (Abcam, #ab239686) were used to capture isolated human plasma EVs according to the manufacturer's instructions. Following the incubation, the beads were washed and stained for CD63 (Proteintech, #67605-1-Ig) and LDHA (Proteintech, #19987-1-AP) antibodies (1:400 dilution) for 1.5 hrs at room temperature. Beads were then washed and incubated with secondary antibodies, followed by washing and resuspending in 300 µl of staining buffer and run immediately on a flow cytometer. After the incubation with secondary antibodies, the samples were analysed on a flow cytometer. Beads only and IgG were used as the negative controls.

## Immunoblotting

Immunoblotting was performed following standard protocol[3,79]. Briefly, cells were lysed using RIPA buffer (Thermo Fisher, #89901) supplemented with a protease inhibitor cocktail (Millipore, #11697498001). Samples were applied to SDS-PAGE gels (GenScript, #M00652) and blotted onto a nitrocellulose membrane (Bio-Rad, #1704270). Membranes were then incubated with primary antibodies (1:1,000 dilution) overnight at 4 °C, and then were incubated with HRP-conjugated secondary antibodies (1:1,000 dilution; CST, #7076 S and #7074 S) for 1 hr at room temperature. Signaling was exposed with chemiluminescence (Pierce, #34580 and #34076) using the ChemiDoc MP Imaging System (Bio-Rad, #17001402). Antibodies were purchased from the indicated companies, including Vinculin (EMD Millipore, #05-386), β-actin (Sigma, #A3854), LDHA (CST, #2012), CCL7 (Biorbyt, #ORB256344), P-ERK (CST, #4370), ERK (CST, #4695), YAP1 (CST, #14074 S), P-STAT6 (CST, #56554 S), STAT6 (R&D Systems, #AF2167SP), P-STAT3 (CST, #9145 S), STAT3 (CST, #9139 S), STK33 (CST, #95343 S), AKT (CST, #4685), P-AKT (CST, #4060), CD63 (Abclonal, #A5271), ALIX (Abclonal, #A2215), and Calnexin (Abclonal, #A15631). Each assay was repeated at least 3 times.

## Quantitative real-time PCR (RT-qPCR)

Cells were detached with trypsin (Gibco, #25300-054) and pelleted. RNA was isolated using the RNeasy Mini Kit (Qiagen, #74106), and then reverse-transcribed into cDNA with the All-In-One 5X RT MasterMix (Applied Biological Materials, #G592). PCR was performed using the SYBR Green PCR Master Mix (Bio-Rad, #1725275). Approximately 10 ng of template was used per PCR reaction. The expression of each gene was quantified using the ΔΔCt method and normalized to the housekeeping gene (e.g., ACTB or GAPDH). PCR was run using the CFX Connect Real-Time PCR Detection System (Bio-Rad, #1855201). Primers are listed in Table S7.

## Immunofluorescence and immunohistochemistry

Immunofluorescence and immunohistochemistry were performed using a standard protocol[3,79]. In brief, a pressure cooker (Bio SB, #7008) was used for antigen retrieval using antigen unmasking solution (Vector Laboratories, #H-3301) at 95 °C for 30 min. After blocking with 10% goat serum for 1 h, slides were incubated with primary antibodies (1:200 dilution) overnight at 4 °C. Slides were then washed with PBS and incubated with secondary antibodies (Invitrogen and CST, 1:500) for 1 hr at room temperature. Slides were then counterstained with DAPI/anti-fade mounting medium (Vector Laboratories, #H-1200-10) for immunofluorescence staining or developed with DAB Quanto (Epredia, #TA125QHDX) followed by hematoxylin for immunohistochemistry staining. Primary antibodies against following proteins were used: STAT3 (CST, #9139 S), YAP1 (CST, #14074 S), Ki67 (Thermo Fisher, #RM-9106-S0), cleaved caspase 3 (CST, #9661 S), F4/80 antibody (CST, #70076 S), LDHA (CST, #2012), and Mac-2 (Biolegend, #125403).

## Hematoxylin and Eosin (H&E) staining

Staining was performed using the H&E staining kit (Abcam, #ab245880) according to a standard protocol. Briefly, tumor sections were incubated with hematoxylin, Mayer's (Lillie's Modification) for 5 min after washing two times in distilled water, and then incubated with the Bluing Reagent and Eosin Y Solution (Modified Alcoholic) for15 sec and 3 min, respectively. The images of tumor sections were captured using TissueFAXS in the Center for Advanced Microscopy (CAM) at Northwestern University.

## ChIP-PCR

ChIP-PCR was performed using the commercial Pierce™ Magnetic CHIP kit (ThermoFisher, #26157) according to the manufacturer's instructions. Briefly, control and sh*Ldha* CT2A cells were cross-linked using 1% PFA (10 min), and then reactions were quenched with glycine (5 min) at room temperature. Cells were lysed with ChIP lysis buffer for 30 min on ice. Chromatin fragmentation was performed using a sonicator. Solubilized chromatin was then incubated with a mixture of YAP1 antibodies (CST, #14074 S) or STAT3 (CST, #9139 S) antibodies and Dynabeads (Life Technologies) overnight. Immune complexes were washed with RIPA buffer three times, once with RIPA-500, and once with LiCl wash buffer. Elution and reverse-crosslinking were performed in direct elution buffer containing proteinase K (20 mg/ml) at 65 °C overnight. Eluted DNA was used to perform qPCR. The primers were designed according to the E-box of mouse *Ccl2* and *Ccl7* genes. Primers are listed in Table S7.

## Microarray and RNA-Seq analysis

The gene expression in human glioblastoma was analyzed using gene-profiling data from the microarray TCGA datasets. For RNA-seq analysis, the total RNA of control and FX11-treated CT2A cells was extracted using RNeasy Kit (Qiagen, #74034). RNA-seq was performed by the Genomics Facility at the University of Chicago. Oligo-dT based library was prepared and samples were sequenced by novaseq 6000 sequencer. Raw data were mapped to the mouse genome. The transcriptome of each gene in control and FX11-treated groups was further quantified. GSEA was used for pathway analyses based on differentially expressed genes of these two groups.

## scRNA-seq data analysis

For the analysis of scRNA-seq data from glioblastoma patient tumors, low-quality cells with detected genes <500, and mitochondrial genes > 20% were removed. Batch effected was removed by CCA-based integration method in Seurat[80]. Both canonical genes and cluster differential genes were used to identify the cell types. scRNA-seq data from the Gene Expression Omnibus (GEO) repository, GSE84465[41], were used to perform unsupervised sub-clustering for macrophages and microglia [CD68 and CX3CR1 were selected as the positive control for TAM (macrophage + microglia) and microglia clustering, respectively]. The expression of LDHA in macrophages, microglia, and other tumor cells was investigated. Next, the scRNA-seq data of GEO, GSE131928[34], were used to analyze the connection among glioblastoma cell glycolysis (including glycolysis signature and key enzymes) and myeloid cells (including macrophage, monocyte, microglia, and DC) in patient tumors. The average expression of each gene and gene signature was represented by color (low to high was shown as blue to red).

## Computational analysis of human glioblastoma datasets

For analysis of human glioblastoma data, we downloaded the microarray gene expression and survival data of TCGA Agilent-4502A dataset (IDH-mutant glioblastoma tumors were excluded, $n = 478$) from Glio-Vis: http://gliovis.bioinfo.cnio.es/. Using this dataset, analyses for the correlation between genes and/or gene signatures, survival, and GSEA of interesting gene and/or gene signatures were performed as we reported previously[3,12,13]. For survival and GSEA analyses, the gene and/or gene signature expression of high and low was defined as top 25% and bottom 25%, respectively. The immune score data of IDH-WT TCGA glioblastoma tumors (Agilent-4502A) was downloaded from the ESTIMATE website: https://bioinformatics.mdanderson.org/estimate/[31]. The immune score dataset was overlapped with gene expression dataset (both are from the TCGA Agilent-4502A dataset) and common patient samples ($n = 300$) were used for correlation analysis, where the correlation between immune score and metabolism/glycolysis signature in corresponding tumors were performed using the Pearson's correlation test.

## Patient samples

Peripheral blood plasma from meningioma ($n = 15$) and glioblastoma ($n = 54$) patients, and tumor samples ($n = 30$) from surgically resected IDH-WT glioblastomas were collected at the Northwestern Central Nervous System Tissue Bank (NSTB) under the institutional review board protocol STU00095863. All patients were diagnosed according to the WHO diagnostic criteria by neuropathologist Dr. Craig Horbinski. The informed consent for research was obtained from the patients. Detailed patient information is provided in Table S8. For control plasma ($n = 15$), we used commercially available anonymized and de-identified, which were isolated from healthy human blood (Solomon Park Research Laboratories, #4345).

## Statistical analysis

Statistical analysis was conducted using GraphPad Prism 9 (GraphPad Software). Data distribution was assumed to be normal, but this was not formally tested. Data collection and analysis were not performed blind to the conditions of the experiments. We did not perform randomization of study participants or samples within each group because not relevant/needed for this study. Statistical analyses were performed with Student $t$-tests for comparisons between two groups or one-way ANOVA tests for comparisons among groups. Data was represented as mean ± SD or SEM as indicated. The survival and correlation analyses in brain cancer datasets (including TCGA dataset) and animal models were performed using the Log-rank (Mantel-Cox) test and the Pearson's correlation test, respectively (GraphPad Prism 9). $P < 0.05$ was considered significant.

## Reporting summary

Further information on research design is available in the Nature Portfolio Reporting Summary linked to this article.

## Data availability

The RNA-Seq dataset generated during this study has been deposited in the GEO repository and the accession number is GSE216070. The previous published scRNA-seq data of GEO, GSE84465[41] and GSE131928[34] were used in this paper. Source data are provided with this paper.

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

## Acknowledgements

We thank Drs. Samuel D. Rabkin, Hideho Okada, Frederick Lang, and Jian Hu for providing 005 GSC, SB28, GSC272, and QPP7 GSC, respectively. This work was supported in part by NIH R00 CA240896 (P.C.), NIH R01 NS124594 (P.C.), NIH R01NS127824 (P.C.), DoD Career Development Award W81XWH-21-1-0380 (P.C.), Cancer Research Foundation Young Investigator Award (P.C.), Lynn Sage Scholar Award (P.C.), and American Cancer Society Institutional Research Grant IRG-21-144-27 (P.C.). Imaging work was performed at the Northwestern University Center for Advanced Microscopy generously supported by NCI CCSG P30 CA060553 awarded to the Robert H Lurie Comprehensive Cancer Center. Exosomes nanopartical analysis was performed in the Analytical bioNanoTechnology Core Facility of the Simpson Querrey Institute for BioNanotechnology at Northwestern University supported by the Soft and Hybrid Nanotechnology Experimental (SHyNE) Resource (NSF ECCS-2025633).

## Author contributions

F.K. performed most of the experiments and analyzed data. Y.L. and W.H.H. performed single-cell sequencing and RNA-sequencing data analysis. H.A., L.P., and M.D. helped with mouse work and ChIP-PCR assay. K.F. and R.G.W isolated and cultured primary human BMDMs. L.K.B. and J.M. helped with the seahorse assay. K.M. and C.M.H. provided help with human patient samples. D.A.W. and M.S.L. contributed to comment on this research and manuscript. P.C. designed the project, analyzed data, and wrote the manuscript. All authors participated in editing the manuscript.

## Competing interests

The authors declare no competing interests.
