## [Peer Review File · Nature Communications]

LDHA-regulated tumor-macrophage symbiosis promotes glioblastoma progressionEditorial Note: This manuscript has been previously reviewed at another journal that is not operating a transparent peer review scheme. This document only contains reviewer comments and rebuttal letters for versions considered at *Nature Communications*.

REVIEWER COMMENTS

Reviewer #1 (Remarks to the Author):

The authors have extensively addressed the concerns raised in the initial submission. Although the figures are dense and difficult to follow in many sub panels, and the reader/reviewer needs to spend a lot of time to get the details, there is improvement overall. The major scientific questions have been addressed with revised figures and increased data.

Reviewer #3 (Remarks to the Author):

The authors have satisfactorily addressed all my comments.

Reviewer #4 (Remarks to the Author):

NCOMMS-23-33297-T

The study by Khan et al. investigates the role of LDH on macrophage migration and the interplay with Glioblastoma cells. The authors claim that some of the observed findings are mediated by EVs, however that part lacks a fundamental EV work up and the plasma EV data should be neglected as the authors used a precipitation kit, which is more or less worthless for EV analysis. Any data out of these kits is biased in so many ways (see multiple reviews by Hendrix et al., Nieuwland et al.)

Major concerns:

- in general the term GBM is outdated and should be revised to glioblastoma throughout the whole manuscript (figures and text)

- the clinical relevance of their findings relies on Figure 1 A. yet it is not stated if this association of OS with metabolism signature is really an independent factor. The authors need to show, what are other covariates. Age, Karnofsky, Extent of Resection, MGMT promotor methylation, treatment, etc. to show that this rather weak association is really a factor for overall survival or if it is biased by any of the aforementioned covariates. And if this does not hold true the translational relevance is weak or even absent.

- the authors have shown most of their data in GL261 and CT2A cells. GL261 is one of the oldest GBM syngeneic cell lines and due to the very high immunogenicity of this cell line does not really reflect the microenvironmental changes of a GBM. CT2A is better in this regard but then there is really only 1 murine line they show their results in. They say they did something in 005 mice, but where is that data? I can't find it in the really overcrowded figures, which need to be simplified to read this manuscript.

- the EM pictures in 5e do not really show anything to me. Where is the lipid bilayer structure? These are only roundish structure of who know what?

- DiD staining of EVs is not optimal. DiD can form aggregates, and the color uptake may only be an artifact. There are multiple other EV marking system, some even link the marker to EV structures in cell cultures, which are far better. (CD63 linked RFP/GFP, Palm constructs, etc.)

- Also the DiD uptake looks far too bright for a physiological EV uptake. How many EVs were used? This is not stated.

- Why did the authors use a precipitation kit for the patient serum samples (figure 7)? It is now well known, that all these EV precipitation kits are more or less worthless and the results can not be trusted. The authors need to show their findings using either UC or SEC. The later should be used if scalability is a problem for the authors.

- Claiming that they have more LDHA in the EV fraction of GBM patient is very weak if the

data comes from WB analysis. how many EVs were loaded? Is it just a coprecipitation artifact? Which is very likely.

- Would be better to do this with either a bead based EV assay, or by any small particle flow, to show that the LDHA is actually on EVs in plasma samples.

Reviewer #5 (Remarks to the Author):

Although I still have some concerns regarding the revised manuscript, I can appreciate the effort of the authors. The current format is ok for publication and the interpretation can be judged later on by readers.

Re: NCOMMS-23-33297-T

Reviewer #4

The study by Khan et al. investigates the role of LDH on macrophage migration and the interplay with Glioblastoma cells. The authors claim that some of the observed findings are mediated by EVs, however that part lacks a fundamental EV work up and the plasma EV data should be neglected as the authors used a precipitation kit, which is more or less worthless for EV analysis. Any data out of these kits is biased in so many ways (see multiple reviews by Hendrix et al., Nieuwland et al.)

Major concerns:

- in general the term GBM is outdated and should be revised to glioblastoma throughout the whole manuscript (figures and text)

We thank the review for pointing this out and have now changed “GBM” to “glioblastoma” throughout the manuscript.

- the clinical relevance of their findings relies on Figure 1 A. yet it is not stated if this association of OS with metabolism signature is really an independent factor. The authors need to show, what are other covariates. Age, Karnofsky, Extend of Resection, MGMT promotor methylation, treatment, etc. to show that this rather weak association is really a factor for overall survival or if it is biased by any of the aforementioned covariates. And if this does not hold true the translational relevance is weak or even absent.

As suggested by the Reviewer, we have now performed these additional analyses with results showing that the metabolism signature was enriched in mesenchymal and IDH1-WT glioblastoma, but not related to the status of tumor recurrence, gender and MGMT promotor methylation (**Extended Data Fig. 1a-e**). These findings exactly support our manuscript’s translational relevance given mesenchymal and IDH-WT glioblastomas have poor survival and high infiltration of macrophages ^{1, 2, 3, 4}.

- the authors have shown most of their data in gl261 and CT2A cells. GL261 is one of the oldest GBM syngeneic cell lines and due to the very high immunogenicity of this cell line does not really reflect the microenvironmental changes of a GBM. CT2A is better in this regard but then there is really only 1 murine line they show their results in. They say they did something in 005 mice, but where is that data? I can’t find it in the really overcrowded figures, which need to be simplified to read this manuscript.

We appreciate the Reviewer’s comment. Indeed, we have *in vivo* data with 005 GSC model (originally shown in **Extended Data Fig. 13f** and **Extended Data Fig. 14o, p**). We are sorry for this confusion and apologize for not making it clearer. However, we hope that the Reviewer can appreciate our effort in putting such a huge number of data (24 figures in total) together in a single manuscript. To address the Reviewer’s comment, we have now moved this survival data to the

main figure (**Fig. 6e**). In addition to CT2A, 005 GSC and GL261 mouse models, we also included human GSC272 PDX model throughout in the manuscript.

- the EM pictures in 5e do not really show anything to me. Where is the lipid bilayer structure? These are only roundish structure of who know what?

We totally agree with the Reviewer and have now removed this data in the revised manuscript.

- DiD staining of EVs is not optimal. DiD can form aggregates, and the color uptake may only be an artifact. There are multiple other EV marking system, some even link the marker to EV structures in cell cultures, which are far better. (CD63 linked RFP/GFP, Palm constructs, etc.).
- Also the DiD uptake looks far too bright for a physiological EV uptake. How many EVs were used? This is not stated.

DiD staining is a classic and effective method for labelling EVs, especially for *in vitro* studies. We cannot agree with the Reviewer more that CD63 linked RFP/GFP and Palm constructs are better than DiD staining. To follow the Reviewer's suggestion, we have ordered the Exosome Cyto-Tracer, pCT-CD63-GFP (#CYTO120-VA-1, System Biosciences) and tried our best to develop this system into macrophages. Unfortunately, the item was on a backorder and then we also realized the difficulties of developing this system into sh*Ldha* macrophages since both lentivirus vectors (pCT-CD63-GFP and shRNAs) are based on puromycin as the selection approach. We also tried GFP sorting to obtain CD63-GFP⁺ shC and sh*Ldha* macrophages. However, the transfection efficiency was low and few GFP⁺ cells were sorted (we are continuing to culture and monitor these cells). These challenges made us unable to obtain enough CD63-GFP⁺ cells for isolating exosomes within a limited revision timeline of 4 weeks.

We agree with the Reviewer that the DiD uptake looks far too bright for a physiological EV uptake. This is largely due to high intensity confocal exposure. To address this concern, we re-did these experiments and provided new images to support the conclusion that glioblastoma cells exhibit equal uptake efficiency for EVs from control macrophages, as well as CT2A EMφ and GL261 EMφ expressing control and *Ldha* shRNAs (**Extended Data Fig. 10b, d**). Moreover, we stated that 500 ng EVs were used in the Figure legends.

- Why did the authors use a precipitation kit for the patient serum samples (figure 7)? It is now well known, that all these EV precipitation kits are more or less worthless and the results can not be trusted. The authors need to show their findings using either UC or SEC. The later should be used if scalability is a problem for the authors.

This is an excellent suggestion. We have now used the SmartSEC Single EV Isolation System to isolate EVs from plasma samples of healthy controls and glioblastoma patients.

- Claiming that they have more LDHA in the EV fraction of GBM patient is very weak if the data comes from WB analysis. how many EVs were loaded? Is it just a coprecipitation artifact? Which is very likely.
- Would be better to do this with either a bead based EV assay, or by any small particle flow, to show that the LDHA is actually on EVs in plasma samples.

Thank you for these helpful suggestions and comments. We have now performed beads-based flow cytometry and found that LDHA in CD63⁺ EVs of glioblastoma patient plasma was significantly higher than that in healthy controls (**Fig. 7l, m**).

References for rebuttal:

1. Chen, P. *et al.* Symbiotic Macrophage-Glioma Cell Interactions Reveal Synthetic Lethality in PTEN-Null Glioma. *Cancer Cell* **35**, 868-884 e866 (2019).
2. Wang, Q.H. *et al.* Tumor Evolution of Glioma-Intrinsic Gene Expression Subtypes Associates with Immunological Changes in the Microenvironment. *Cancer Cell* **32**, 42-+ (2017).
3. Khan, F. *et al.* Macrophages and microglia in glioblastoma: heterogeneity, plasticity, and therapy. *J Clin Invest* **133** (2023).
4. Hartmann, C. *et al.* Patients with IDH1 wild type anaplastic astrocytomas exhibit worse prognosis than IDH1-mutated glioblastomas, and IDH1 mutation status accounts for the unfavorable prognostic effect of higher age: implications for classification of gliomas. *Acta Neuropathol* **120**, 707-718 (2010).

REVIEWER COMMENTS

Reviewer #4 (Remarks to the Author):

The authors revised their study and added more data to this manuscript which in my opinion is overcrowded already. in general they did a huge effort but-

the authors did not understand my concern about their Figure 1A

if you run a correlation analysis on survival time (OS) and your newly identity factor (in your case your metabolism signature) is associated with poor survival you need to prove this in an univariate analysis and than you need to show in a multivariate analysis that this new factor is significantly associated with survival and independent of other co-factors.

now a major concern arises when i see the extended figure 1A in which the authors show IDHmutated gliomas. These are NOT glioblastoma and should be taken out of this study. So the authors need to revise their clinical cohort and exclude all IDHmutated samples. They actually already show in their extended figure 1A that IDHmutated tumors have a lower score, which might explain why these patients live longer, since IDH mutated gliomas live longer anyways. Also we need to see here Karnofsky, extend of resection and treatment choice (combined radiotherapy, best supportive care, etc.) to draw any clinical relevant conclusion.

In regards to their EV analysis it is good that they redid their work with the SEC isolation kit but now they need to show that they really isolated EVs, which they should do with an EM but this time showing real bilipid, maybe even cup shaped EVs.

Looking at their flow EV data it is hard for me to understand how they did this. Because (in their setting CD63+ EVS and then LDHA) usually someone would take beads that can pull CD63+ positive EVs and then stain with CD63 and LDHA. I don't understand why the authors used plain polystyrene beads. How are the EVs specifically been captured by the beads? The flow data lacks a multitude of controls. like antibody controls and so on.

Re: NCOMMS-23-33297A

Reviewer #4

The authors revised their study and added more data to this manuscript which in my opinion is overcrowded already. In general they did a huge effort but the authors did not understand my concern about their Figure 1A. If you run a correlation analysis on survival time (OS) and your newly identified factor (in your case your metabolism signature) is associated with poor survival you need to prove this in an univariate analysis and then you need to show in a multivariate analysis that this new factor is significantly associated with survival and independent of other co-factors. Now a major concern arises when I see the extended figure 1A in which the authors show IDHmutated gliomas. These are NOT glioblastoma and should be taken out of this study. So the authors need to revise their clinical cohort and exclude all IDHmutated samples. They actually already show in their extended figure 1A that IDHmutated tumors have a lower score, which might explain why these patients live longer, since IDH mutated gliomas live longer anyways. Also we need to see here Karnofsky, extent of resection and treatment choice (combined radiotherapy, best supportive care, etc.) to draw any clinically relevant conclusion.

We thank the Reviewer for her/his appreciation regarding our huge efforts on this manuscript. We totally agree with the Reviewer that this manuscript is overcrowded with 24 figures in total. However, we respect these 5 Reviewers for asking so many questions and comments to improve this manuscript.

We are sorry for the misunderstanding of the comment raised by the Reviewer on Figure 1A. To follow this comment, we have now removed the data regarding the correlation between metabolism signature and survival. On the other hand, we performed the Kaplan-Meier curve analysis and confirmed that the metabolism signature negatively correlated with patient survival (**Fig. 1a**). We provided a statement in the Figure legend and Methods sections that only IDH-WT glioblastoma tumors were included for analysis.

In regards to their EV analysis it is good that they redid their work with the SEC isolation kit but now they need to show that they really isolated EVs, which they should do with an EM but this time showing real bilipid, maybe even cup shaped EVs.

We thank the Reviewer for these additional comments. We have now performed EM and nanoparticle tracking analysis for these isolated EVs and confirmed their identity (**Fig. 7I and Extended Data Fig. 16m**).

Looking at their flow EV data it is hard for me to understand how they did this. Because (in their setting CD63+ EVs and then LDHA) usually someone would take beads that can pull CD63+ positive EVs and then stain with CD63 and LDHA. I don't understand why the authors used plain polystyrene beads. How are the EVs specifically been captured by the beads? The flow data lacks a multitude of controls. like antibody controls and so on.

Thank you for these comments. Indeed, a large body of studies have shown that polystyrene beads can capture isolated EVs. However, we totally agree with the Reviewer that using the CD63 EV capture beads is better. We have now performed such experiments and confirmed that LDHA in glioblastoma patient plasma EVs was significantly higher than that from healthy controls (**Fig. 7m, n** and **Extended Data Fig. 16n**). On the other hand, we have now provided beads only and IgG negative controls in our analysis (**Fig. 7m, n**).

REVIEWER COMMENTS

Reviewer #4 (Remarks to the Author):

The authors redid their survival analysis:

eventhough the authors claim that they removed the IDH-mt gliomas I wonder why in Panel Fig.1B the plot stayed the same. Because in the TCGA often there is an overlap if IDH-wt and IDH-mt tumors. So they need to be very careful here.

On the other hand their clinical data (comparing bottom and top 25%) is relatively weak and thus that this is clinically relevant is not backed up by the data they present

In regards to their EV data:

- The Bead gating in their extended figures is strange, as this reviewer does not understand why they did not solely gate on the bead population which is really clear and visible in their first gating panel

- Also the EM pictures that they show are really uncommon and this reviewer wonders if they really have EVs here, however the rest: NTA and CD63 pull is somewhat convincing, if they show that their claims hold true if they redo their bead gating strategy.

We redid the gating as suggested by the Reviewer and provided detailed information in the Methods section regarding how the analysis was performed to generate Fig. 1b, which specific TCGA dataset (IDH-WT tumors and number of cases) was used, and what type of analysis was performed (e.g., how cases were selected).

Reviewer #4

The authors redid their survival analysis:

eventhough the authors claim that they removed the IDH-mt gliomas, I wonder why in Panel Fig.1B the plot stayed the same. Because in the TCGA often there is an overlap if IDH-wt and IDH-mt tumors. So they need to be very careful here.

On the other hand their clinical data (comparing bottom and top 25%) is relatively weak and thus that this is clinically relevant is not backed up by the data they present.

We thank the Reviewer for these further comments. We have not included IDH-mutant tumors even in the first version of the manuscript and stated it in the figure legend. That's why the Fig. 1b stayed the same as previous versions. As guided by the editor, we have now provided clear statements in the Methods section. It is common to define high and low as top 25% and bottom 25%, respectively, for survival analysis.

In regards to their EV data:

- The Bead gating in their extended figures is strange, as this reviewer does not understand why they did not solely gate on the bead population which is really clear and visible in their first gating panel
- Also the EM pictures that they show are really uncommon and this reviewer wonders if they really have EVs here, however the rest: NTA and CD63 pull is somewhat convincing, if they show that their claims hold true if they redo their bead gating strategy.

We appreciate the Reviewer's comment that the results of NTA and CD63 pull are convincing. We followed the Reviewer's suggestion to redo the gating (**Extended Data Fig. 16n**) and confirmed that LDHA in glioblastoma patient plasma EVs was significantly higher than that from healthy controls (**Fig. 7m, n**). It should be noted that the samples used for flow cytometry analysis are only beads (but not from tumor tissues that may contain multiple cell populations). Therefore, we tried to include the majority of beads for the following flow cytometry analysis in the previous version of manuscript.

REVIEWERS' COMMENTS

Reviewer #4 (Remarks to the Author):

All my minor points were addressed

Re: MS# NCOMMS-23-33297C

Response to the Reviewers:

Reviewer #4:

All my minor points were addressed.

We thank the Reviewer for appreciation of our efforts to revise this manuscript.